# Tunable exciton valley-pseudospin orders in moiré superlattices

Richen Xiong[1], Samuel L. Brantly[1], Kaixiang Su[1], Jacob H. Nie[1], Zihan Zhang[1], Rounak Banerjee[2], Hayley Ruddick[2], Kenji Watanabe[3], Takashi Taniguchi[4], Seth Ariel Tongay[2], Cenke Xu[1] & Chenhao Jin[1] ✉

Excitons in two-dimensional (2D) semiconductors have offered an attractive platform for optoelectronic and valleytronic devices. Further realizations of correlated phases of excitons promise device concepts not possible in the single particle picture. Here we report tunable exciton "spin" orders in $WSe_2/WS_2$ moiré superlattices. We find evidence of an in-plane ($xy$) order of exciton "spin"—here, valley pseudospin—around exciton filling $\nu_{ex} = 1$, which strongly suppresses the out-of-plane "spin" polarization. Upon increasing $\nu_{ex}$ or applying a small magnetic field of ~10 mT, it transitions into an out-of-plane ferromagnetic (FM-$z$) spin order that spontaneously enhances the "spin" polarization, i.e., the circular helicity of emission light is higher than the excitation. The phase diagram is qualitatively captured by a spin-1/2 Bose–Hubbard model and is distinct from the fermion case. Our study paves the way for engineering exotic phases of matter from correlated spinor bosons, opening the door to a host of unconventional quantum devices.

Properties and applications of correlated systems can go beyond those attainable within the single particle framework. For example, magnets with spontaneous electron spin ordering form the cornerstone of spintronics[1]; superconductors[2] and phase-change transistors[3] offer potential routes to next-generation electronics. Moiré superlattices recently emerged as a powerful playground to engineer correlated phenomena[4]. Along with the strong light-matter interaction and unique optical selection rules[5,6], correlated excitons in semiconducting moiré systems hold promises for novel applications in photonics and valleytronics[7–9]. However, while various magnetic orderings and correlated phases of electrons are reported, such as correlated insulator[10–14], superconductivity[15–17], intrinsic[18,19] or exciton-mediated ferromagnetism[20], and fractional quantum anomalous Hall states[21–24]; correlated phases of excitons remain unexplored until very recently[25–28], and exciton "magnets" with "spin" orders have not been demonstrated.

Here we observe an intriguing phase diagram of interlayer-exciton "spin" orders in $WSe_2/WS_2$ moiré superlattices near one exciton per lattice site. Spin-up and spin-down exciton "spins" correspond to (and hereafter refer to) the $K$ and $K'$ valleys—two degenerate but inequivalent corners of the hexagonal Brillouin zone—and are related by time-reversal symmetry[5]. Similar to magnetism of real spins, exchange interaction between excitons of different valley pseudospins can lead to spontaneous ordering of the valley degree of freedom. For example, an out-of-plane ferromagnetic (FM-$z$) spin order polarizes spins to the same out-of-plane direction, which, in the exciton context, corresponds to a state where excitons are spontaneously polarized to the same valley; On the other hand, an in-plane spin is a coherent superposition of spin-up and spin-down. Therefore, excitons in an in-plane ($xy$) order are each a superposition between the two valleys of equal amplitude[29,30]. To probe exciton "spin" order, we use a pump–probe spectroscopy[25] that isolates the low energy excitations of the system (Fig. 1a, see Methods: "Pump–probe spectroscopy"). Similar to electrical capacitance measurements[31,32], the DC pump light controls the background exciton density and maintains the quasi-equilibrium state, while the AC probe light injects a small perturbation of extra excitons and isolates their responses through lock-in detection. Owing to the optical selection rules in transition metal dichalcogenides, left- and

[1]Department of Physics, University of California at Santa Barbara, Santa Barbara, CA, USA. [2]School for Engineering of Matter, Transport, and Energy, Arizona State University, Tempe, AZ, USA. [3]Research Center for Functional Materials, National Institute for Materials Science, Tsukuba, Japan. [4]International Center for Materials Nanoarchitectonics, National Institute for Materials Science, Tsukuba, Japan. ✉e-mail: jinchenhao@ucsb.edu

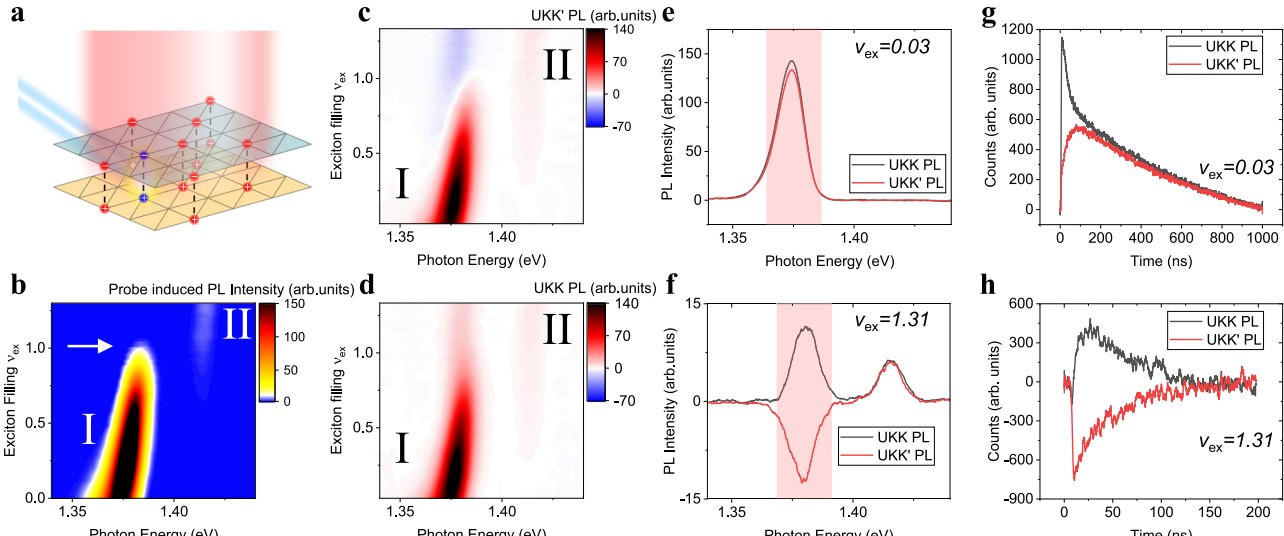

**Fig. 1 | Spin-1/2 Bose–Hubbard model. a** Schematics of our pump–probe spectroscopy on a type-II hetero-bilayer WSe$_2$/WS$_2$. The background interlayer-exciton density (red) is controlled by the pump light and the charge density is kept at zero. The probe light injects an extra interlayer-exciton (blue), whose response is isolated through lock-in detection. **b** Exciton-filling dependence of probe-induced PL spectrum using unpolarized pump and probe light. A sudden jump of exciton chemical potential at $v_{ex} = 1$ is observed (white arrow), indicating an incompressible state of excitons. Low (high) energy emission peak is denoted as peak I (II). **c**, **d** Polarization-resolved probe-induced PL spectra as a function of exciton filling. A linear pump light is used to generate equal numbers of $K$ and $K'$ valley excitons in the background, while an LCP probe light selectively excites extra $K$ valley excitons. $K'$ valley (**c**) or $K$ valley (**d**) PL response induced by the probe light is collected separately. UKK (UKK') refers to pump injecting unpolarized excitons, probe injecting $K$ valley excitons and PL detecting $K$ ($K'$) valley excitons. **e**, **f** Linecuts of (**c**, **d**) at low (**e**) and high (**f**) exciton density. **g**, **h** Probe-induced TRPL signals from peak I at low (**g**) and high (**h**) exciton density. A constant background signal from the CW linear pump light is subtracted. Pink boxes in (**e**, **f**) denote the spectral filter used to isolate peak I response. The negative signals in UKK' configuration (**h**) indicate that $K$ excitons selectively form doublons with $K'$ excitons.

right-circularly polarized (LCP and RCP) light selectively couple to spin-up and spin-down excitons[5,8,33]. This allows us to independently control spin of the background excitons and the extra injected excitons through polarization of the pump and probe light, respectively; and obtain spin-resolved system response from polarization-resolved photoluminescence (PL) detection. We can thereby directly create and probe low-energy spin excitations of the system.

## Results

### Spin-1/2 Bose–Hubbard model

Figure 1b shows the pump–probe PL spectrum of a 0-degree-aligned WSe$_2$/WS$_2$ moiré device D1 with unpolarized pump, probe light, and PL detection (the electron density is kept at 0 throughout this study). Unpolarized light couples to the total population of excitons[5,33,34]. The measurement therefore directly obtains the energy to remove one exciton from the system, i.e., its chemical potential. A sudden jump of exciton chemical potential is observed at one exciton per moiré site ($v_{ex} = 1$, $n_{ex} = 1.9 \times 10^{12}$ cm$^{-2}$, see Methods: "Calibration of background exciton density"), corresponding to a bosonic-correlated insulator state[25]. The low and high energy peaks, labeled peak I and II, correspond to PL emissions from a singly occupied site and a doublon site (site with two excitons). Their energy shift of ~30 meV provides a direct measurement of exciton-exciton on-site repulsion and indicates the strong correlation between excitons[25,35].

We then switch experimental configurations to inspect spin excitations of excitons. The minimum model to account for exciton "spins" is a two-component Bose–Hubbard model[36], given by

$$H = \sum_{<i,j>,\alpha} -t b_{i,\alpha}^{\dagger} b_{j,\alpha} + h.c. + \sum_{i,\alpha} U(n_{i,\alpha} - 1/2)^2 + \sum_i V n_{i,1} n_{i,2} \quad (1)$$

Here the $\alpha = 1, 2$ label $K$ and $K'$ valley pseudospin of excitons, $t$ is hopping between nearest neighboring sites, and the interactions between excitons consist of intra-species repulsion $U$ and inter-species

repulsion $V$. To establish such model and separately determine $U$ and $V$, we use linearly polarized pump light to generate equal population of two "spins" in the background, an LCP probe light to selectively inject extra $K$ valley ("spin"-up) excitons, and monitor "spin"-resolved responses by separately collecting RCP ($K'$ valley) and LCP ($K$ valley) PL from the probe light only (Fig. 1c, d). The $K'$ and $K$ responses are rather similar at $v_{ex} < 1$ (Fig. 1e), which can be understood in the single exciton picture from a short valley lifetime that quickly relaxes valley polarization. We directly capture such relaxation process by time-resolved pump–probe PL measurements (Fig. 1g). The pulsed probe light selectively injects $K$ excitons at time zero, and the $K'$ response remains unchanged. Afterwards, the valley polarization quickly disappears over time, resulting in similar overall responses from the two valleys (Fig. 1e).

In contrast, the two valleys' responses become dramatically different above $v_{ex} = 1$ (Fig. 1f). Most strikingly, their responses have opposite signs for peak I. The negative $K'$ response indicates that adding extra $K$ excitons will decrease the number of singly occupied $K'$ sites. Such behavior is incompatible with the single exciton picture where adding $K$ excitons always increase both $K$ and $K'$ exciton populations[34,37] and is instead a unique consequence of exciton correlation. Our observation can be naturally understood from the Bose–Hubbard model: the $K$ excitons injected by the probe form doublons at $v_{ex} > 1$, which will decrease the number of singly occupied sites by converting them into doublon sites. The decrease in $K'$ sites therefore indicates that $K$ excitons selectively form doublon sites with $K'$ excitons (Supplementary Fig. 2a). This is further confirmed by the perfect valley balance in doublon emission (peak II) regardless of experimental configuration (Supplementary Fig. 2b), which requires $K$ and $K'$ excitons to be symmetric within any doublons.

We note that while peak I is expected to disappear at $v_{ex} \geq 1$ in exciton chemical potential measurement, as is observed in Fig. 1b, it should persist in spin-resolved measurements until $v_{ex} = 2$ (Fig. 1c, d) due to the spin-selective formation of doublons discussed above. This can also be understood by the cancellation of peak I signals when

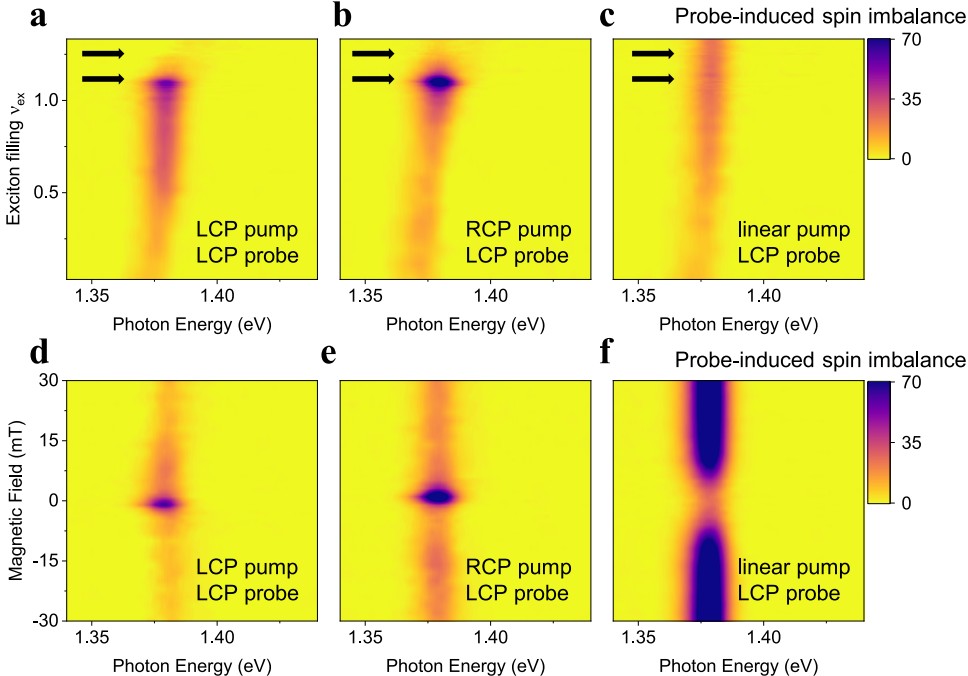

**Fig. 2 | Inter-site spin-dependent exciton interaction. a–c** Probe-induced spin imbalance (arbitrary units) as a function of background exciton fillings for LCP (**a**), RCP (**b**), and linear (**c**) pump, respectively. For both LCP and RCP pump, a sharp enhancement of signals at $v_{ex}$ ~ 1.1 followed by a quick drop at $v_{ex}$ ~ 1.2 are observed (black arrows). In contrast, these features are missing under linear pump, indicating rapidly changing exciton spin interaction when doping slightly away from the correlated insulator state. **d–f** Evolution of probe-induced spin imbalance (arbitrary units) at $v_{ex}$ ~ 1.1 under out-of-plane magnetic field $B_z$ for LCP (**d**), RCP (**e**), and linear (**f**) pump, respectively. The signals show sensitive and symmetric change under a small $|B_z|$ ~ 5 mT. All the measurements are performed at base temperature 3 K unless specified.

adding up the two spin channels at $v_{ex} \geq 1$ (Fig. 1c, d), which confirms that the probe light cannot add more single-occupied sites and can only create doublons. Meanwhile, peak II should ideally only emerge at $v_{ex} \geq 1$. The weak peak II features observed at $v_{ex} < 1$ in Fig. 1c, d are mainly due to the large intensity of the pulsed probe light used here for time-resolved measurements, as well as the spatial inhomogeneity in the exciton density (see Methods: "Discussions on doublon emission"). These effects could be reduced by using a weaker probe and a more homogeneous sample, such as in ref. 25. A systematic study on probe intensity and sample homogeneity would help further quantify these effects.

Our results thus unambiguously establish a "spin"-dependent on-site repulsion between excitons. The ~30 meV jump of exciton chemical potential at $v_{ex} = 1$ (Fig. 1b) corresponds to the opposite-"spin" repulsion $V$, while the same-"spin" repulsion $U$ is much greater than $V$. Consequently, doublons only form by two excitons of opposite "spins" like electrons in a Fermi–Hubbard model, which offers a rare realization of spin-½ Bose–Hubbard model.

**Inter-site spin-dependent interactions between excitons**

Next, we investigate inter-site spin interactions that may lead to spin orders. We vary the pump light polarization and keep the probe LCP. Different pump polarization maintains background excitons of different spins, while the LCP probe light always injects spin-up excitons. Any difference in the measured probe response can therefore directly reflect spin-dependent interaction between excitons. Figure 2a–c shows the probe-induced spin imbalance (the difference between $K$ and $K'$ emission induced by the probe light) as a function of exciton fillings for LCP, RCP, and linear pump, respectively. See Supplementary Fig. 4 for results on another device D2. Peak II always shows zero spin-imbalance signal, as expected for doublon emission (Supplementary Fig. 2). Peak I is insensitive to pump polarization at low exciton density, suggesting negligible spin interaction effects. At increasing exciton

density, in contrast, the signals vary dramatically with pump polarization. Under both LCP and RCP pump (Fig. 2a, b), we observe a sharp signal enhancement at $v_{ex}$ ~ 1.1 followed by a quick drop at $v_{ex} > 1.2$ (black arrows). Both features are absent in the linear pump case (Fig. 2c), indicating their origin from spin interactions. While the strongest spin interaction in the system is the on-site AFM exchange, it cannot account for the symmetric behaviors between the LCP and RCP pump or the sensitive filling dependence (see Methods: "Spin-1/2 Bose–Hubbard model"). These features therefore indicate rapidly changing inter-site spin interactions when doping slightly away from a bosonic-correlated insulator.

We first investigate the feature at $v_{ex} = 1.1$ and monitor its evolution under an out-of-plane magnetic field $B_z$. Figure 2d–f shows the magnetic field-dependent spin imbalance spectra for fixed $v_{ex} = 1.1$ and different pump polarization. Surprisingly, the probe-induced spin imbalance under CP pumps or linear pump are either suppressed or enhanced by an order of magnitude, respectively, upon applying a tiny magnetic field of 5 mT. Such sensitive magnetic field dependence and low saturation field (~20 mT) of exciton spin polarization have not been reported before[38–41], and generally indicates adjacent phase transitions with strong spin fluctuations[42].

To further confirm the phase transition, we also perform pump–only PL measurement. Such measurement collects PL from all background excitons in the system and is therefore less sensitive to spin interactions than the pump–probe measurement. However, a spin order will affect not only low energy excitations but all excitons in the system and should therefore be observable in such measurements. Figure 3a shows example spectra of $K$ and $K'$ PL at $v_{ex} = 1.39$ with RCP pump, from which we obtain the PL raw helicity $\eta_{PL} = \frac{I_{K,PL} - I_{K',PL}}{I_{K,PL} + I_{K',PL}}$. $I_{K,PL}$ and $I_{K',PL}$ are the PL emission intensity from $K$ and $K'$ excitons (peak I in Fig. 3a), which are proportional to the number of singly occupied $K$ and $K'$ sites, respectively. Figure 3b summarizes $\eta_{PL}$ under different pump polarization and exciton fillings. To reveal spin orders, we introduce

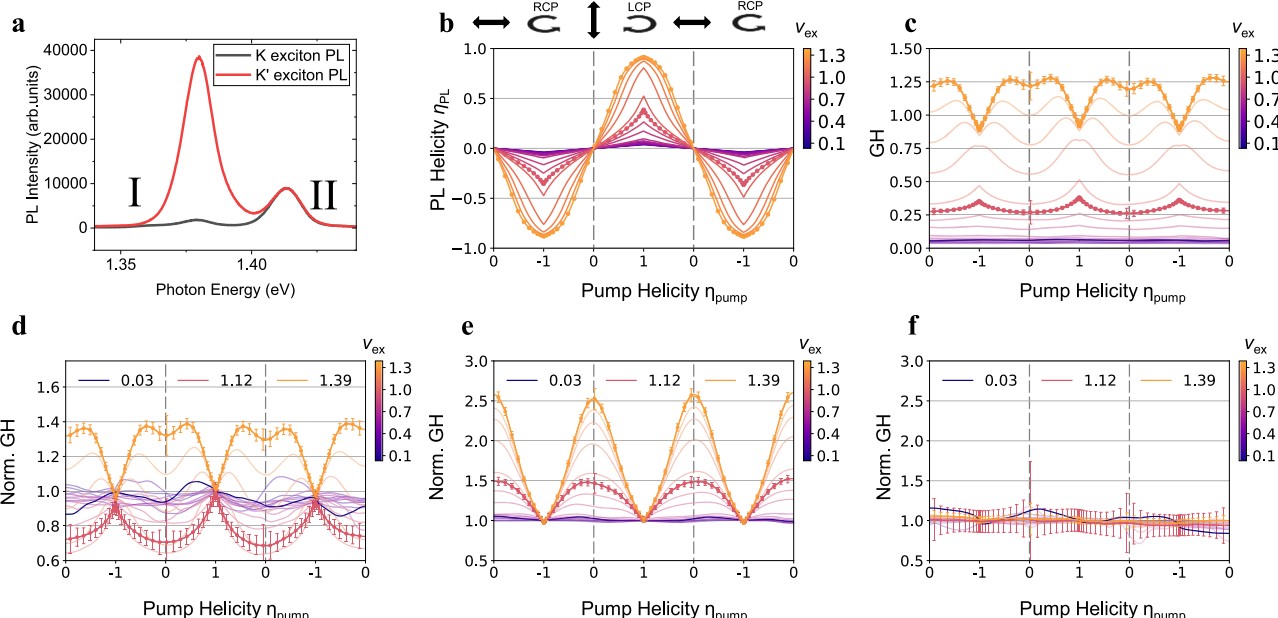

**Fig. 3 | Evidence of exciton spin orders from generalized helicity (GH).**
**a** Pump–only PL from K (black) and K′ (red) valley with RCP pump at $v_{ex} = 1.39$. Peak I shows a large PL helicity while peak II has no PL helicity. **b**–**d** PL raw helicity $\eta_{PL}$ (**b**), generalized helicity (GH, defined as $\eta_{PL}/\eta_{pump}$) (**c**) and normalized GH (**d**) of peak I under different pump helicity $\eta_{pump}$ and exciton fillings $v_{ex}$. GH does not depend on $\eta_{pump}$ at low exciton density, consistent with the single-particle picture with no spin order. In contrast, GH becomes "Λ" shape at $v_{ex}$ - 1.1 and quickly transitions into "V" shape at $v_{ex} > 1.25$, indicating existence of mean-field from exciton spin order.

**e** Normalized GH at $B_z = -30$ mT (see $B_z = 30$ mT in Supplementary Fig. 7a). The "Λ" shape at $v_{ex}$ - 1.1 becomes "V" shape under ±30 mT field. The sensitive and symmetric $B_z$ dependence echoes with the pump–probe measurement results.
**f** Normalized GH at 60 K. GH remains flat over the whole exciton filling range. GH and normalized GH at $v_{ex} = 0.03, 1.12, 1.39$ are highlighted in (**c–f**). Data are shown as lines and symbols for $v_{ex} = 1.12$ and 1.39; and only lines are shown at other fillings for visual clarity. Error bars represent standard deviation in PL helicity, GH and normalized GH (**b–f**, see Methods: "Data analysis").

generalized helicity GH = $\eta_{PL}/\eta_{pump}$, where $\eta_{pump}$ is the helicity of pump light (see Methods: "Data analysis" and Supplementary Fig. 9). In the case of no spin order, the LCP and RCP components of pump light should independently contribute to PL emission, and therefore GH will be a constant over pump polarization. A spin order, on the other hand, generates a mean field that depends on all exciton spins in the system and thus on the pump polarization. Consequently, GH will change with pump helicity $\eta_{pump}$.

Figure 3c, d shows GH and normalized GH over pump polarization (helicity) at different exciton fillings. Since GH is not well-defined at $\eta_{pump} = 0$ (linear pump), only data at $|\eta_{pump}| > 0.02$ are obtained experimentally (symbols); and GH at $\eta_{pump} = 0$ can be extrapolated from the limit of $\eta_{pump} \to 0$ (see Methods: "Data analysis"). At low exciton density GH is indeed a constant over pump polarization. In contrast, GH becomes a "Λ" shape at $v_{ex} = 1.1$ and quickly transitions into a "V" shape at $v_{ex} > 1.25$, echoing the two features in the pump–probe PL spectra (Fig. 2a, b). When we further apply an out-of-plane magnetic field, the constant GH at low exciton filling remains intact (Fig. 3e). The "Λ" shape GH at $v_{ex} = 1.1$, on the other hand, changes dramatically and becomes a "V" shape under both ±30 mT field (see Supplementary Fig. 7a for data at 30 mT). Such sensitive and symmetric magnetic field dependence is consistent with the pump–probe results and again signifies an adjacent phase transition. We have also measured normalized GH at 60 K as a reference (Fig. 3f), which is flat over the whole exciton density range. This further confirms the origin of nontrivial shapes in GH from exciton spin orders.

**Tunable transient exciton spin orders**

To unravel the nature of the exciton spin orders, we performed detailed magnetic field dependence at $v_{ex} = 1.1$ and 1.3. Figure 4a, b show the PL raw helicity and GH at $v_{ex} = 1.1$ from 0 to 30 mT. Intriguingly, the GH at $B_z > 20$ mT exceeds unity in a wide pump polarization range. A GH > 1 means that the spin polarization of the system is higher

than the pump. This cannot be explained by field-induced symmetry breaking between the two spins, which would favor one spin over the other and lead to asymmetric GH between RCP and LCP pump. In contrast, the observed GH is symmetric against $\eta_{pump} = 0$ and exceeds unity on both sides (Fig. 4b). If excitons in the system were not decaying over time—equivalently, all PL emissions are re-absorbed by the system—the system would keep amplifying the spin polarization. A tiny initial spin-up/down injection would then eventually develop into a close to fully spin-up/down state. Such spontaneous spin polarization is the hallmark of an FM-z order. On the other hand, because here excitons are in a quasi-equilibrium between decaying and pumping, any system memory is lost over the exciton lifetime and there should not be hysteresis. Hence all orders identified experimentally are of transient nature at the timescale of exciton decay.

Our observation indicates a transient FM-z order of excitons at $v_{ex} = 1.1$ and magnetic field $B_z > 20$ mT. The zero-field state at $v_{ex} = 1.1$ is more exotic. Phenomenologically, it shows opposite behaviors from the high field FM-z order in both pump–probe and GH measurements (Figs. 2a–c and 4b), indicating distinct exciton spin states. Its extremely sensitive magnetic field dependence and low saturation field (~20 mT) is particularly surprising. The most well-known effect from an out-of-plane magnetic field is the Zeeman splitting that lifts the degeneracy between the two exciton spins. Similarly, the dominant effect of magnetic field on excitons is an energy splitting between the two valleys, termed "valley Zeeman effect"[38,39], with a g-factor of 4 in monolayer TMD. However, the splitting should be <-0.1 meV under 10 mT[34], which is too small compared to the expected energy scale of spin interaction (-1 meV, see Methods: "Discussions on magnetic field dependence"). In addition, applying positive and negative $B_z$ should lead to opposite Zeeman splitting. Instead, in our experiment they have mostly symmetric effects and eventually result in a transition into the same FM-z order (Figs. 2d–f, 3e and Supplementary Fig. 7a). Both observations exclude a simple linear coupling between the Zeeman

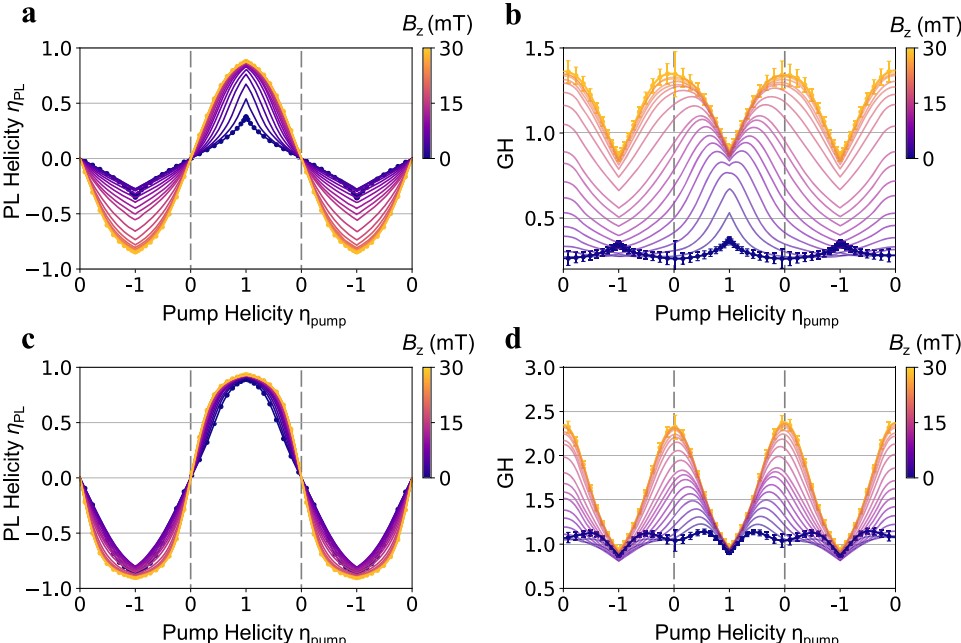

**Fig. 4 | Tunable exciton spin orders with magnetic field and exciton filling.**
**a**, **b** Magnetic field dependence of PL raw helicity $\eta_{PL}$ (**a**) and GH (**b**) at $v_{ex} = 1.1$ from 0 to 30 mT. The GH transforms from a "Λ" shape to a "V" shape and saturates at ~20 mT, suggesting a phase transition. At high field, GH exceeds 1 in a wide range of pump helicity, indicating spontaneous increase of spin polarization, which is the hallmark of an FM-$z$ spin order. **c**, **d** Same as (**a**, **b**) for $v_{ex} = 1.3$. The zero field GH is similar to the high field GH at $v_{ex} = 1.1$, signifying an FM-$z$ spin order at zero field. The "V" shape GH is further enhanced by $B_z$. Data are shown as lines and symbols for $B_z = 0$ and 30 mT; and only lines are shown at other magnetic fields for visual clarity. Error bars represent standard deviation in PL helicity and GH. Error bars in (**a**–**c**) are smaller than the symbol size.

field and the order parameter and suggest a finite in-plane component in the zero-field order, i.e., an $xy$ order (For more discussions, see Methods: "$xy$ order").

Indeed, such phase transition can fully explain our experimental observations. Under linear pump, the $xy$ spin order creates an in-plane mean field, which will efficiently mix up and down spins and suppress spin polarization. On the other hand, spins under CP pumps are initialized to be along the $z$ direction and the in-plane mean field is weaker. We therefore expect a stronger suppression of spin polarization near linear pump and a weaker suppression near CP pumps, i.e., a "Λ" shape GH (Fig. 4b). The high field FM-$z$ order, on the contrary, amplifies spin polarization. This leads to a sharp rise of $\eta_{PL}$ with $\eta_{pump}$ and a large GH > 1 when $|\eta_{pump}|$ is small. At large $|\eta_{pump}|$, $\eta_{PL}$ saturates since it cannot exceed 1 (Fig. 4a); and GH is always smaller than 1 when $|\eta_{pump}| = 1$ (CP pumps). We therefore expect a "V" shape GH as observed experimentally (Fig. 4b). The transition region between the $xy$ and FM-$z$ orders at intermediate field is more complicated, where GH becomes asymmetric between positive and negative $\eta_{pump}$ or $B_z$ (Fig. 4b). This indicates extrinsic symmetry breaking between the two spins by the magnetic field and thus no well-defined order (see Methods: "Discussions on magnetic field dependence").

We now turn to the feature at $v_{ex} > 1.25$. Its zero field behaviors are qualitatively similar to the high field behaviors at $v_{ex} = 1.1$ in all measurements: pump–probe measurement (Fig. 2) shows a stronger spin imbalance signal under linear pump compared to CP pump; GH shows a "V" shape with GH > 1 over a wide range of pump helicity. Upon applying an out-of-plane magnetic field, these behaviors are qualitatively unchanged and quantitatively enhanced. For example, GH is enhanced to a giant value of 2.3 near linear pump (Fig. 4d), corresponding to a rapid increase and saturation of spin polarization as $\eta_{pump}$ increases that can be clearly seen in the PL raw helicity (Fig. 4c). These results provide strong evidence that at $v_{ex} > 1.25$ the system is already in an FM-$z$ order without magnetic field, i.e., suggesting a filling-controlled transition between $xy$ to FM-$z$ order at $v_{ex} \sim 1.25$.

The pump–probe measurement results (Fig. 2) are also naturally explained by the competition between the in-plane and out-of-plane spin interactions of excitons. At $v_{ex} = 1.1$, the dominant in-plane spin interactions under a linear pump rapidly quench out-of-plane spin imbalances, while a CP pump reduces such quench by forcing exciton spins to be out-of-plane. We therefore observe stronger probe-induced spin imbalance signals under both LCP and RCP pumps compared to the linear pump case. Upon increasing exciton filling and/or magnetic field $B_z$, the FM-$z$ spin interactions dominate and enhance spin imbalances under linear pump as the system is not fully polarized. Once all spins are polarized under CP pump, the system is not susceptible to spin excitations anymore and the probe-induced spin imbalance becomes vanishingly small.

We next measure the temperature dependence of these orders. Figure 5a, b shows normalized GH at $v_{ex} = 1.1$ and 1.3, respectively, for temperatures from 3 to 60 K. The "Λ" shape GH at $v_{ex} = 1.1$ and "V" shape GH at $v_{ex} = 1.3$ disappear at around 35 and 50 K, respectively, indicating melting of the associated spin orders. To quantify the temperature dependence, we define $\Delta_{GH} = \frac{GH(CP\ pump) - GH(linear\ pump)}{GH(CP\ pump)}$. A positive and negative $\Delta_{GH}$ correspond to a "Λ" shape and "V" shape GH and indicates $xy$- and $z$- spin order, respectively. Figure 5c, d shows the phase diagram of $\Delta_{GH}$ at 0 and −30 mT (see Supplementary Fig. 7b for 30 mT data). We also mark regions where GH (linear pump) >1 with dotted texture, which is the hallmark of an FM-$z$ order. At zero field, the system first enters an $xy$ order upon increasing exciton filling to $v_{ex} \sim 1$ and then transitions into an FM-$z$ order at $v_{ex} \sim 1.25$. At magnetic field of −30 mT the $xy$ order is suppressed, and the FM-$z$ order is favored over the filling range of $v_{ex} > 1.1$. Its melting temperature keeps increasing with the exciton filling.

## Discussion

Our observations provide strong evidence of phase transitions from a (transient) $xy$ order to FM-$z$ order driven by both exciton filling and magnetic field. This can be naturally understood in a spin-½

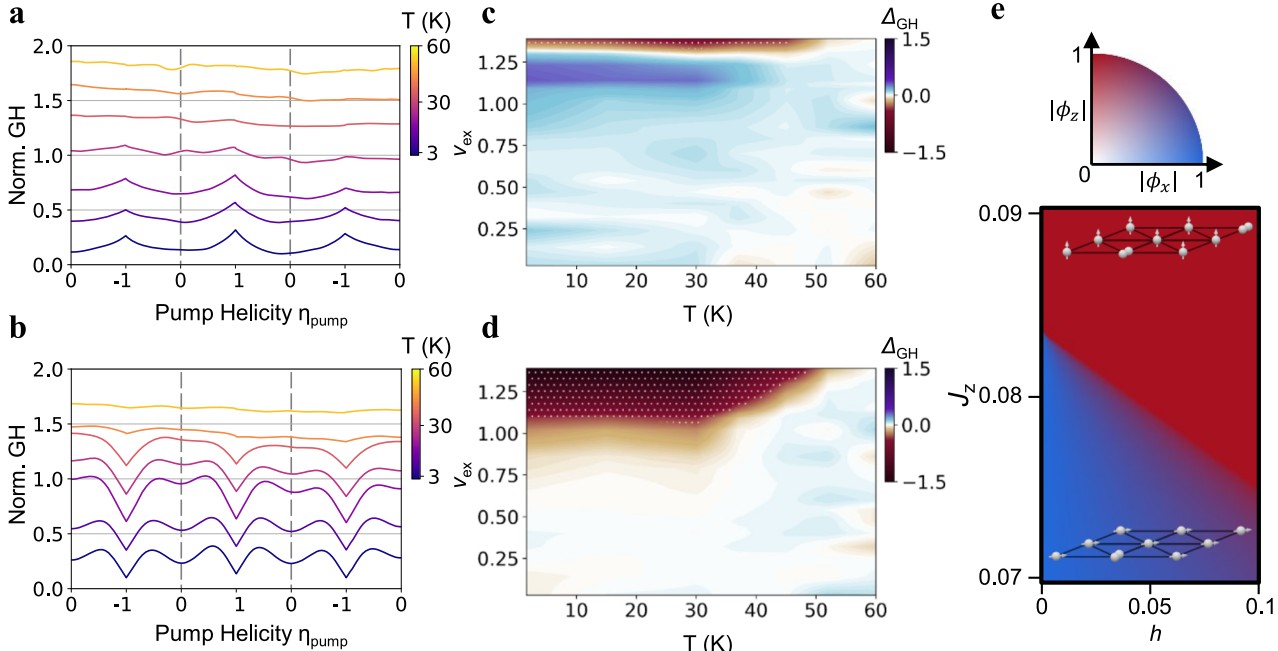

**Fig. 5 | Phase diagram of exciton spin orders. a, b** Temperature dependence of normalized GH at $\nu_{ex} = 1.1$ (**a**) and $\nu_{ex} = 1.3$ (**b**). The "Λ" shape and "V" shape feature melt at around 35 K and 50 K, respectively. **c, d** Phase diagrams of $\Delta_{GH}$ at $B_z = 0$ mT (**c**) and −30 mT (**d**). A positive (negative) $\Delta_{GH}$ corresponds to a "Λ" ("V") shape GH and indicates $xy$ ($z$) spin order. White dotted texture marks regions with GH > 1 at linear pump, which is the hallmark of an FM-$z$ spin order. **e** Theoretical phase diagram from a phenomenological spin ½ XXZ model, where $h$ is the out-of-plane magnetic field and $J_z$ is $z$-direction exchange interaction. The in-plane exchange $J_\perp$ is fixed to be 1/6. $\phi_x$ ($\phi_z$) is the expectation value of $S^x$ ($S^z$). The color represents orientation of the order parameter and the opacity represents its amplitude. Effects from adding extra excitons to a $\nu_{ex} = 1$ correlated insulator are captured by $J_z$ that increases with exciton filling. A transition from the FM-$xy$ to FM-$z$ order is expected upon both increasing $J_z$ (exciton filling) and magnetic field, which is consistent with our experimental observations.

Bose–Hubbard model from competitions between the super-exchange effect and Nagaoka-type kinetic ferromagnetism[19,43]. Our pump–probe measurements establish WSe₂/WS₂ moiré superlattice as a spin-½ Bose–Hubbard model, which has been predicted to host a ground state of FM-$xy$ order at $\nu_{ex} = 1$ as the virtual hopping of bosons gives rise to an FM in-plane super-exchange interaction $J_\perp$[36,44]. Upon further doping, the kinetic energy of extra bosons would favor an FM-$z$ order, similar to Nagaoka ferromagnetism in Fermi–Hubbard model[19,43,45]. We reveal the essential physics near $\nu_{ex} = 1$ using a phenomenological spin-½ XXZ model on a triangular lattice, where the effect of adding excitons to the $\nu_{ex} = 1$ correlated insulator is captured by a $z$-exchange interaction $J_z$ that increases with doping (see Methods: "Theoretical phase diagram" and Supplementary Note 1). Figure 5e shows the phase diagram predicted by this phenomenological model, which matches well with the experimental one. A transition from FM-$xy$ to FM-$z$ order is expected with increasing doping (and $J_z$). In addition, the system is very sensitive to a Zeeman field $B_z$ near the transition; and a weak $B_z$ would favor the FM-$z$ over the FM-$xy$ order. Intriguingly, both the FM-$xy$ and FM-$z$ orders are unique consequences of Bose–Einstein statistics–it is well-established that the super-exchange interaction in a Fermi–Hubbard model is antiferromagnetic (AFM) along all directions[45,46], and the Nagaoka FM is also isotropic instead of favoring the $z$ direction[43,45]. While the phenomenological model here captures the salient features from the experiment, more quantitative theoretical studies are warranted to fully understand the exciton behaviors observed. The recent success of calculating moiré excitons from GW-BSE[47] could allow direct prediction of the exciton hopping in the Bose–Hubbard model. Furthermore, additional effects not captured by the Bose–Hubbard model, such as exciton dissociation and decoherence between the two valleys, could also contribute to the exotic exciton responses. Our results provide a valuable experimental reference for future theoretical studies.

Our study establishes semiconducting moiré superlattices as an intriguing platform to realize exotic states of excitons, which will also open up novel device concepts in photonics and quantum information science. For example, an FM-$z$ order not only stores but also amplifies valley polarization, which may serve as a cornerstone of memory and error correction code[48,49]. The extremely sensitive magnetic field and exciton filling dependence are consequences of phase transitions and go beyond the single-particle limit, which may enable efficient light source control and optical gates similar to phase-change transistors in electronics[3,50]. In addition, the two-component Bose–Hubbard model can potentially support a plethora of much more exotic phases beyond the ferromagnetic orders. For instance, it was shown numerically that a supersolid can be realized in the effective XXZ spin-1/2 model with one boson per-site[44]. Away from the vicinity of one boson per site, the two flavors of hard-core bosons can be mapped to an SU(4) spin model (with anisotropies, see Supplementary Note 1), which is a system that has attracted enormous interest in the past few decades and hosts various intriguing phases[51–57].

## Methods

### Device fabrication and characterization

The dual-gated WSe₂/WS₂ devices were made by layer-by-layer dry transfer method[58]. Polarization-resolved second harmonic generation (SHG) was used to determine the crystalline angles between monolayer WSe₂ and WS₂ before stacking. hBN flakes with a thickness of around 20 nm were used as the gate dielectrics and few-layer graphite flakes were used as the gates and contact electrode. The whole stack was then released to a 90 nm Si/SiO₂ substrate with pre-patterned Au contacts. Supplementary Fig. 1a shows an optical image of 0-degree aligned WSe₂/WS₂ device D1. The twist angles were measured by SHG to be within 0 ± 0.5°, limited by experimental uncertainties. Supplementary Fig. 1b, c shows the gate-dependent absorption and PL characterization of the moiré bilayer at 3 K. At charge neutrality (~−0.05 V), the PL features a single peak at 1.375 eV from interlayer exciton emission, while the absorption shows three peaks from moiré intralayer

excitons. At $\nu_e = 1$ and $\nu_h = 1$ (white arrows), the emission peak blueshifts suddenly and the absorption peaks show a kink, indicating the emergence of correlated insulator of charges. All measurements are performed at a base temperature of 3 K in 0°-aligned WSe$_2$/WS$_2$ device D1 unless specified.

## Pump–probe spectroscopy

The samples were mounted in a closed-cycle cryostat (Quantum Design, OptiCool). A continuous wave 660 nm diode laser was used as the pump light with beam size of around 100 μm². The large pump beam size ensures a homogeneous intensity in the center region that is inspected by the probe beam. A pulsed 680 nm light from a supercontinuum laser (YSL Photonics, 300 ps pulse duration, variable repetition rate) were used as the probe light. The beam size of probe light was around 4 μm². The probe intensity was kept below 30 nW/μm², while the pump intensity ranged from 0 to 3 μW/μm². To isolate the response from probe-created excitons, the probe light was modulated by an optical chopper at frequency of 10 Hz. The signal was detected by a liquid-nitrogen-cooled CCD camera coupled with a spectrometer (Princeton Instruments), which was externally triggered at 20 Hz and phase locked to the chopper. The spectra with and without the probe light were thereby obtained, and their difference gives the signals from the probe light only (see Fig. 1b for an example). To help isolate the probe light response, we also implement a spatial filter at a conjugate image plane of the sample, which only allows light from the probe-covered region to go through. For polarization-resolved measurements, the polarization of pump/probe/PL is controlled by broadband polarizers and half-wave plates. In time-resolved PL measurements (TRPL), the signals are collected by an avalanche photodiode (MPD PDM series) and analyzed by a time-correlated single photon counting module (ID Quantique ID1000). Since the pump light is CW while the probe light is pulsed, their contribution can be directly separated in the time domain and no AC modulation is needed. We thereby directly track the system's dynamical response to extra transient excitons at certain background exciton filling.

## Calibration of background exciton density

We precisely calibrate the exciton density and filling at each pump intensity through time-resolved PL measurements. This is done in two steps. We first perform time-resolved PL (TRPL) measurement using the CW pump light as excitation light (Supplementary Fig. 10a). PL emission rate is a constant over time, as expected from CW excitation. This allows us to determine the emission rate at each pump intensity (Supplementary Fig. 10c). Next, we establish the relation between emission rate and exciton density by replacing the CW pump with a pulsed pump light (300 ps pulse duration, 1 MHz repetition rate) with the same wavelength (660 nm) and beam profile. All other experimental configurations are also kept identical. Supplementary Fig. 10b shows the time-resolved PL using the pulsed excitation light of different pulse fluences. The decay dynamics changes with pulse fluence but is always much slower than the instrumental response function (IRF, Supplementary Fig. 10b inset). Therefore, the emission rate immediately after time-zero corresponds to the exciton density created by the pulsed excitation light without any relaxation, which can be directly obtained from the pulse fluence.

The above procedure allows us to reliably determine exciton density without complications from the exciton lifetime or relaxation dynamics. Since the system reaches quasi-equilibrium in a short time (<1 ps), each measured emission rate uniquely corresponds to one exciton density at quasi-equilibrium, whether the excitation light is CW or pulsed. For example, at charge neutrality we identify $\nu_{ex} = 1$ at pump intensity of 0.406 μW/μm², which corresponds to an emission rate of 562 (Supplementary Fig. 10c). The same emission rate is achieved by the pulsed pump light with fluence $F = 0.25$ J/m² immediately after time zero (Supplementary Fig. 10d). The exciton density is directly obtained

from the pulse fluence though $n_{ex} = \alpha F/(h\nu) = (1.9 \pm 0.2) \times 10^{12}$ cm$^{-2}$, where $F$ is the pulse fluence, $\alpha = (0.023 \pm 0.002)$ is absorption of WSe$_2$ at 660 nm using its dielectric function and considering the multi-layer structure of our device, $h\nu$ is the photon energy of 660 nm light. $n_{ex}$ matches well with the expected exciton density $n_0 = \frac{2}{\sqrt{3}a_M^2} = 1.9 \times 10^{12}$ cm$^{-2}$ at $\nu_{ex} = 1$, where $a_M \sim 8$ nm is the moiré periodicity considering 4% lattice mismatch and 0-degree twist angle. We calibrate the exciton density at all pump intensity and polarization following the above procedure.

## Data analysis

To ensure the reliability of GH = $\eta_{PL}/\eta_{pump}$, we carefully calibrate the uncertainties in both $\eta_{pump}$ and $\eta_{PL}$. In our experiment, the pump light goes through a half-wave plate (HWP) and a quarter-wave plate (QWP) before impinging on the sample. The pump helicity $\eta_{pump}$ is controlled by the HWP angle $\theta$ and should ideally follow a simple sine function of $\eta_{pump} = \sin(\pi \frac{\theta - \theta_0}{45°})$. On the other hand, imperfect optics and/or alignment may result in deviation from such relation. To calibrate the uncertainty in $\eta_{pump}$, we directly measure the LCP and RCP components in the sample-reflected pump light using identical experimental configuration as measuring the LCP and RCP PL, as shown in Supplementary Fig. 9a. This allows us to determine $\theta_0$ when the LCP and RCP components are equal. The extracted $\eta_{pump}$ (Supplementary Fig. 9b) shows a near-perfect match with the ideal sine relation (gray curve) and a relative standard deviation $\Delta\eta_{pump}/\eta_{pump} < 2\%$.

To calibrate the uncertainty in $\eta_{PL}$, we measure $\eta_{PL}$ twice under identical experimental configurations at each exciton filling and extract the deviation between the two measurements. Supplementary Fig. 9c shows the results for three representative exciton fillings $\nu_{ex} = 0.02$, 1.12, and 1.39, from which we obtain a standard deviation $\Delta\eta_{PL}$ of 0.21%, 0.17%, and 0.17%. Such a small uncertainty corresponds to an error bar smaller than the symbol size in PL raw helicity (Fig. 3b and Supplementary Fig. 9c). On the other hand, the uncertainty in GH will be dramatically amplified at small $|\eta_{pump}|$ since GH = $\eta_{PL}/\eta_{pump}$, and GH becomes nominally ill-defined at $\eta_{pump} = 0$. We therefore extrapolate GH at $\eta_{pump} = 0$ from the limit of $\eta_{pump} \to 0$. As exemplified in Supplementary Fig. 9d, the uncertainty in GH becomes reasonably small (<5%) when $|\eta_{pump}| > 0.05$ (outside green shaded region), where the GH curve is already flat with pump polarization. This indicates that GH has a well-defined value in the $\eta_{pump} \to 0$ limit, and our extrapolation is reliable. Another way to understand the reliability of GH at small $|\eta_{pump}|$ is that it is simply the slope between $\eta_{PL}$ and $\eta_{pump}$. As one can directly see in the PL raw helicity (Fig. 3b), $\eta_{PL}$ has a well-defined slope near $|\eta_{pump}| = 0$.

## Spin-½ Bose–Hubbard model

To account for the two species of excitons related by time-reversal symmetry, the simplest form of the Bose–Hubbard model reads

$$H = \sum_{<i,j>,\alpha} -t b_{i,\alpha}^\dagger b_{j,\alpha} + h.c. + \sum_{i,\alpha} U(n_{i,\alpha} - 1/2)^2 + \sum_i V n_{i,1} n_{i,2}$$

Here the $\alpha = 1, 2$ label $K$ and $K'$ valley pseudospin of excitons, $t$ is hopping between nearest neighboring sites, and the interactions between excitons consist of intra-species repulsion $U$ and inter-species repulsion $V$. Our flavor-resolved pump–probe results indicate $U > V > 0$, i.e., an on-site AFM interaction. Consequently, doublon sites always form between one $K$ valley and one $K'$ valley exciton, and the chemical potential jump at $\nu_{ex} = 1$ directly measures $V \sim 30$ meV.

On the other hand, the on-site interactions cannot account for the distinctive spin imbalance responses between LCP/RCP pump and linear pump at $\nu_{ex} > 1$ (Fig. 2). The on-site AFM interaction can indeed induce different probe responses between different pump polarizations: an LCP/RCP pump will generate more $K/K'$ background excitons,

which will suppress/assist the formation of doublon sites with the extra $K$ excitons from an LCP probe. Such effect should therefore be opposite under LCP and RCP pump compared to linear pump, which is incompatible with the symmetric behaviors of features in Fig. 2 under both LCP and RCP pumps. In addition, the strength of the on-site interaction effects scales linearly with the doublon site density and should continuously increase with $\nu_{ex}$; while experimental features show sensitive and non-monotonic filling dependence. Last but not least, the effect from the AFM on-site interaction remain largely intact up to 60 K (Supplementary Fig. 5). However, all the observed features disappear at 60 K (Fig. 5 and Supplementary Fig. 6). These pieces of evidence indicate a dominant role of inter-site spin interactions to features in Figs. 2–5.

## Theoretical phase diagram

Inter-site spin interactions naturally emerge in the spin-½ Bose–Hubbard model. One well-established mechanism is the super-exchange effect $J_{SE} \sim t^2/V$ (see Supplementary Note 1 and ref. 36). In a Fermi–Hubbard model, $J_{SE}$ is isotropically AFM. In a spin-½ Bose–Hubbard model, in contrast, its in-plane components $J_{SE,xy}$ are FM. Quantitatively, we estimate $J_{SE,xy}$ to be ~1 meV using typical hopping energy of moiré excitons $t \sim 5$ meV (refs. 7,59) and the measured inter-species on-site interaction energy $V \sim 30$ meV. We can also independently estimate $J_{SE,xy} \sim 1$ meV from the temperature dependence using $6J_{SE,xy} = k_B T_c$, where $T_c = 35$ K is the melting temperature of the $xy$ order. The consistency between two independent methods further confirms the validity of our estimation. The out-of-plane component of the super-exchange effect $J_{SE,z} \sim (2t^2/U \cdot t^2/V)$ can be either FM or AFM[36]. Since $U > V > 0$ guarantees $|J_{SE,z}| < |J_{SE,xy}|$, an FM-$xy$ order is expected at the Mott insulator state of $\nu_{ex} = 1$. On the other hand, further doping of excitons will lead to additional effective FM interaction $J_N$ similar to Nagaoka ferromagnetism in Fermi–Hubbard model[43,45] (see Supplementary Note 1). While $J_N$ is isotropic in the Fermi–Hubbard model with SU(2) spin rotation symmetry, the symmetry is explicitly broken in the spin-½ Bose–Hubbard model when $U \neq V$; and $J_N$ favors FM-$z$. As $J_N$ grows with exciton filling above $\nu_{ex} = 1$, it eventually makes $J_z = J_{SE,z} + J_N > J_{SE,xy}$, leading to a transition between FM-$xy$ to FM-$z$ order. We encapsulate competition between the super-exchange effect and the Nagaoka-type ferromagnetism in a phenomenological XXZ model (see Supplementary Note 1), which successfully captures all salient features of the experimental observation.

The two-orbital Bose–Hubbard model can potentially support a plethora of much more exotic phases beyond the ferromagnetic orders being discussed here. For example, it was shown numerically that a supersolid can be realized in the effective XXZ spin-1/2 model with one boson per-site[44]. In the Supplementary Note 1, we will also briefly discuss the potential exotic phases when we go beyond the vicinity of one boson per site, as two flavors of hard core bosons can be mapped to an SU(4) spin model (with anisotropies), which is a system that has attracted enormous interests in the past few decades as it can be engineered in transition metal oxides with both spin and orbital degrees of freedom, graphene-based moiré systems, as well as cold atoms[51–57]. Various exotic phases of spin systems with exact or approximate SU(4) symmetries have been discussed in literature[60–67].

## Discussions on magnetic field dependence

The observed magnetic field dependence at $\nu_{ex} = 1.1$ (Fig. 4a, b) can be intuitively understood through an interplay between the molecular field and the external field for an $xy$ pseudospin order. At zero external field, the $xy$ pseudospin order leads to finite $|\phi_x|$ and generates an in-plane mean field that aligns pseudospins to in-plane. With a sufficiently large external field, on the other hand, the total field becomes out-of-plane. As a result, $xy$ orders relying on the in-plane mean field are suppressed and $z$ orders are favored. Once the system enters the $z$ order ($|B_z| > 20$ mT), the Zeeman field applied will not significantly

break symmetry between the two pseudospins since the Zeeman energy scale (~0.1 meV at 20 mT) is much smaller than the exchange interaction (~1 meV). Indeed, we observe largely symmetric behaviors under positive and negative field for $|B_z| > 20$ mT (Fig. 3e and Supplementary Fig. 7a). The intermediate field regime ($|B_z| < 20$ mT) shows asymmetric responses for positive and negative $B_z$, indicating external symmetry breaking from the Zeeman field. As a result, there is no well-defined spontaneous symmetry breaking or pseudospin order. The asymmetric behaviors can be qualitatively understood from the fact that $|\phi_x|$ remains finite in this regime. The total field is therefore tilted and can still mix the two pseudospins and induce rapid switching between them, during which the Zeeman splitting favors the flavor with lower energy.

## $xy$ order

The observed PL helicity is always zero at linear pump, indicating zero average out-of-plane "spin". There are two possible scenarios: all sites have in-plane "spin"; or domains of up and down spins. The second scenario applies to the case of $\nu_{ex} > 1.25$, which shows rapid increase of PL helicity with pump helicity and a "V" shape GH, as discussed in the main text. The distinctively different "$\Lambda$" shape GH at $\nu_{ex} \sim 1.1$ thus excludes this scenario and indicate in-plane "spin". In addition, the symmetric and sensitive magnetic field dependence (Fig. 2d–f) indicates a finite in-plane mean field at linear pump. Therefore, the in-plane spin directions cannot be completely random and the system should have at least a finite-range, transient $xy$ order. On the other hand, our observation does not require a long-range $xy$ order and provides no information on the correlation length of the order. A long-range FM-$xy$ order corresponds to a global phase coherence between the two valleys; therefore, PL from the system should be linearly polarized. We did not observe linear helicity in PL (Supplementary Fig. 8), indicating that there is no true long-range FM-$xy$ order. This can be naturally understood as there can only be at most a quasi-long range $xy$ order in 2D at finite temperature[68]; and all orders observed in this work should be of transient nature. Besides intrinsic spin fluctuations and exciton decay, coupling to phonons and other quasiparticles may further introduce decoherence channels acting as a random in-plane magnetic field, thereby limiting the time- and length-scale of the $xy$ order.

## Comparison with spin orders of electrons in moiré systems

Spin orderings of electrons have been widely observed in both TMD[11,19,23,24,69,70] and graphene moiré systems[71,72]. They can be divided into two classes:

1. The spin ordering is associated with band topology. These systems involve two or more orbitals, and the magnetism is intimately connected to topological states such as Chern insulator. Example systems include magic-angle twist bilayer graphene[71], ABC trilayer graphene/hBN[72], AB stacked MoTe$_2$/WSe$_2$ heterobilayers[69] and AA stacked MoTe$_2$ homobilayers[23,24].

2. The underlying band is non-topological. These systems can be described by a single band Fermi–Hubbard model and the spin exchange/orders can be understood from doping a Mott insulator. Examples include frustrated magnetic interaction in WSe$_2$/WS$_2$[11,70] and kinetic magnetism in MoSe$_2$/WS$_2$[19].

The exciton spin orders reported here belong to a category with distinctive different physics. Since the single-particle exciton bands are non-topological, the physics here can be captured by a Hubbard model. Nevertheless, the bosonic nature of excitons gives rise to distinct behaviors from the Bose–Hubbard model, as compared to the Fermion case. In particular, a Bose–Hubbard model predicts in-plane FM super-exchange interaction at $\nu = 1$[36] and out-of-plane FM interaction from Nagaoka-type mechanism at $\nu > 1$; while a Fermi–Hubbard model predicts antiferromagnetic (AFM) super-exchange interaction along all directions at $\nu = 1$[45,46], and the Nagaoka FM is also isotropic

instead of favoring the $z$ direction[43,45]. Our experimental observation of transition between an $xy$ order to an FM-z order is therefore a unique consequence of Bose–Einstein statistics.

## Discussions on doublon emission

Ideally, peak II (emission from doublons) should only appear at $\nu_{ex} \geq 1$ in all measurement configurations. In Fig. 1c, d, peak II emerge below $\nu_{ex} = 1$. This is mainly due to the large intensity of the pulsed probe light we used for time-resolved measurements, which can transiently increase the exciton density by a significant amount to form doublons. Another important practical factor is sample inhomogeneity. Because exciton density depends on exciton lifetime, its spatial inhomogeneity is very sensitive to defects, strain etc. and is expected to be much larger than the charge case. At an average $\nu_{ex} < 1$, there could already be regions in the sample with $\nu_{ex} \geq 1$, which leads to doublon emission. These issues could be addressed or alleviated by using a weaker probe and/or a more homogeneous sample. For example, Fig. R1b uses a continuous wave (CW) probe with much smaller peak intensity. Therefore, peak II emerges only slightly below $\nu_{ex} = 1$.

## Data availability

All data supporting this work in have been deposited in the OSF database with: https://doi.org/10.17605/OSF.IO/XJAHG.

## Code availability

The code used for data analysis in this study have been deposited in the OSF database with: https://doi.org/10.17605/OSF.IO/XJAHG.

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

## Acknowledgements

The work in C.J.'s lab is primarily supported by National Science Foundation (NSF) through a CAREER award DMR-2337606. R.X. acknowledges support from the UC Santa Barbara NSF Quantum Foundry funded via the Q-AMASE-i program under award DMR-1906325. K.W. and T.T. acknowledge support from the JSPS KAKENHI (Grant Numbers 19H05790 and 20H00354). S.T. acknowledges primary support from DOE-SC0020653 (materials synthesis), Applied Materials Inc., NSF CMMI 1825594 (NMR and TEM studies), NSF DMR-1955889 (magnetic measurements), NSF CMMI-1933214, NSF 1904716, NSF 1935994, NSF ECCS 2052527, DMR 2111812, and CMMI 2129412. C.X. is supported by the Simons Investigator program.

## Author contributions

C.J. conceived and supervised the project. R.X., J.H.N. and Z.Z. fabricated the devices. R.X. performed the optical measurements. R.X. and S.L.B. analyzed the data. K.S. and C.X. performed theoretical calculations on the spin model. R.B., H.R. and S.T. grew the $WSe_2$ and $WS_2$ crystals. K.W. and T.T. grew the hBN crystals. C.J. and R.X. wrote the manuscript with the input from all the authors.

## Competing interests

The authors declare no competing interests.
