## [Peer Review File · Nature Communications]

Tunable exciton valley-pseudospin orders in moiré superlatticesReviewers' Comments:

Reviewer #1:

Remarks to the Author:

"Tunable exciton valley pseudospin orders in moiré superlattices" by Xiong et. al. explore the low temperature excitonic properties of a R stacked WSe₂/WS₂ vertical heterostructure. The paper using both pump-probe and pump only PL measurements, focuses on the exploration of excitonic properties link to the 0-degree alignment resulting moiré superlattice. The authors observe that close to filling factor 1 a (xy) valley pseudospin order appears. They demonstrate that an increase of either the filling factor (from 1.1 to 1.25) or magnetic field (20mT) transition the system into a Ferromagnetic Z phase. One may regret that no other type of layer alignment (H stacking) or other filling factor (-1.1 and -1.25 for example) are explored, as it could have given more ground and context to the paper main findings. Nevertheless, the findings of the paper are new and are presented in a clear manner. The resulting interpretation in term of spin 1/2 Bose Hubbard model is convincing and I think that the results clear signature of Bose Einstein physics in excitons would deserve publication in Nature Communication. Nevertheless, there is in my opinion a major problem in the way the paper is presented:

- As stated in the introduction, the paper treats of exciton "magnetic" phase with "spin" order. But the spin is in fact in this case the pseudo spin the valley polarization (K and K'). I understand perfectly that spin and pseudo spin are just quantum numbers and are formally in great part inter-changeable. Nevertheless, this start bothering me as physical effect such as Zeeman, Ferromagnetism, out of plan spin polarization, and the resulting effect of external magnetic field on these phases start to be treated. Those are strongly related to spin (real spin) and concepts that would be trivial when applied to spin are not so when talking about the valley polarization (pseudo spin). I do not think that the findings of the paper or the interpretation are not correct, I just find that this could not only be treated/discussed only through the spin analogy without discussing further the implications more clearly. The paper would gain in quality by being more pedagogic and I would suggest to discuss more thoroughly the results in terms specific to valley polarization, with more identifiable bibliography, equations and schematic representations of the different valley polarized state describe in the paper.

Some other smaller concerns:

- 1- The paper introduction heavily focused on the description of pump probe experiment but the rest of the paper is in fact more focused on the pump only PL experiment, (3 figures versus 1).
- 2- There is a fair amount of information pushed to the Method section at the end of the paper, with 9 different. The paper is made harder to read by the fact that it is not always clear when referred to Method in the paper which part of the Method is concern. I would suggest that some important discussion (From Discussion on the xy order and Theoretical phase diagram) to be transferred into the main text or at the minimum that the part of the method concern to be referred more clearly.
- 3- Line 184 A giant GH value of 2.3 is announced in Figure 3 (d).

Reviewer #2:

Remarks to the Author:

Xiong et al. report experimental studies of inter-layer excitons confined in the moiré potential of a WS₂/WSe₂ bilayer. The authors claim that they have engineered an xy ferromagnetic ordered phase, accessed by mapping the Heisenberg Hamiltonian with the two species Bose-Hubbard (BH) model. Unfortunately I find that the authors claim is not sustained by their experimental observations. As a result I regret not to recommend the work for publication, for the following detailed reasons.

The title of Fig.1 is "spin dependent interactions", however the authors do not report the spin-dependence of on-site interactions in Fig.1. As detailed in Ref.[37] this is essential. To possibly extract the phase diagram of the 2-species BH model, which maps into the Heisenberg model, the magnitude

of the difference between intra- and inter-species interactions is critical. This difference is not quantified here and not explicitly considered. Mott-like insulators crucially depend on the difference between inter and intra-species on-site interactions. In the manuscript a single on-site interaction U is extracted unlike the 3 necessary scattering rates detailed in Ref.[37].

In Fig.1 the authors claim that they measure the chemical potential as a function of the filling of the moiré lattice. However, I am doubtful since in the panel b the amplitude of the probe PL strongly varies with the strength of the pump filling the lattice. Why is that? Why does the probe intensity increase until $\nu=1$ and then suddenly drops? Naively I conclude that the authors also measure the PL radiated by excitons injected by the pump so that their technique is not a pump-probe. As a result, the incompressible nature of the interlayer exciton fluid is not transparent which is worrying since mapping of the xy-model with a 2-species BH system requires experimental conditions deep in the Mott insulating regime, e.g. with exactly one particle per site. At $\nu=1.3$ for example I do not find reasons to consider that excitons realise an insulating phase.

In Fig.1c-g the authors do not justify why the increased probe intensity manifests ordering. For charges (electrons or holes) a similar effect has been reported and correlated to charge ordering through capacitance measurements. Here the same extrapolation can not be done. The sole pump-probe approach does not suffice to reach this conclusion since the authors do not extract the exciton compressibility. For optically active excitons the compressibility is possibly deduced by monitoring PL intensity fluctuations with few sites resolution, see for instance Nature 609 485 (2022). Following the above paragraph I hardly see how the authors can justify that excitons realise any incompressible phase.

In Fig.2 the authors study the PL polarisation as a function of the pump polarisation (with no probe). Fig.2.b shows that the PL has a polarisation that increasingly follows the one from the pump while the moiré filling is increased. At ν around 1.3 Fig.2.b shows that the PL is well polarised by the pump (see Fig.S9) when no magnetic field is applied. Then the authors compute a so-called generalised ellipticity (GH). However I find that GH does not clarify pseudo-spin ordering. I understand that the authors have tried to rely on the best possible calibration, but GH is computed by dividing the PL ellipticity by the one of the pump which diverges for linear pump polarisation. I am then not comfortable with the authors analysis that uses GH as a sort of order parameter. This is not suitable and in fact to my view GH artificially exacerbates differences between pump and PL polarisations. As a matter of fact, in Fig.2.b the polarisation is almost unchanged between $\nu=1.1$ and 1.3 whereas Fig.2.d reports larger differences.

In general I find that the experiments reported in Fig.2 and Fig.3 only reveal an optically induced magnetisation, but I find no evidence for either spin ordering or incompressible exciton phases. For example, at ν around 1.3 the moiré lattice has singly and doubly filled sites. Ref.[37] shows that theoretically this strongly modifies the structure of insulating Mott-like phases. In the manuscript the authors do not address this aspect and the experimental results vary weakly above $\nu=1$ which is very surprising.

In Fig.4 the authors tentatively build a phase diagram based on GH and argue that pseudo-spin ordering persists up to around 30-50K. I regret but I find this conclusion is in contradiction with other measurements of the manuscript. Precisely, Eq.(3) in Ref.[37] shows that xy-order results from the competition between J_{\perp} and h . Here, the authors explore a regime where U is around V_a and V_b , and t_a is around t_b , so that J_z is vanishing and h reduces to hext. xy-ordering theoretically melts when kT is around J_{\perp} , so here around 30-50 K, i.e. 3-5 meV. At the same time, xy ordering melts when h becomes greater than J_{\perp} and the authors find that this occurs at 5-10 mT (Fig.1.h). This comparison leaves orders of magnitude difference, revealing that J_{\perp} is not around 5 meV despite the authors claim. In fact, I am not surprised here since a simple calculation shows that the tunnelling coefficient (t_a and t_b) are of the order of 0.05 meV for a moiré depth estimated around 100 meV. Since the authors measure that U is around 30 meV, J_{\perp} is reduced to around 1 micro-eV so that

xy-ordered phases are far out of experimental reach here. The linewidth of the PL (10-20 meV) further supports my conclusion.

To conclude, Xiong et al. claim that interlayer excitons realise xy-ordered phases in a moiré lattice. This is a very strong claim and to my view the experimental results do not provide a ground to reach such conclusion. I only deduce from their work that they report optically induced magnetism which does not deserve publication in Nat. Comm.

Reviewer #3:

Remarks to the Author:

This manuscript reports on the finding of magnetic orders of excitons in the angle-aligned moire superlattice of WSe₂/WS₂. This is a very interesting follow up of the discovery of exciton insulator in this system by the same group (ref. 21). Using a strong CW pump and weak pulsed probe, the author showed convincing evidences for magnetic orders from the valley pseudo spins, as experimentally revealed by circular polarized PL, of the photo-excited interlayer excitons. The sensitivity of the observed spin orders to very weak external magnetic field along the z-direction suggests the presence of adjacent phase transitions. While these findings are of interest to the broad community working on moire physics, the writing needs to be substantially improved before considerations for publication. Here are issues that require attention.

1) The (valley) spin order from interlayer excitons can be considered as light-induced magnetism. It is strange that the authors failed to discuss relevant publications on the topic, e.g., the paper of Xiaodong Xu and coworkers Nature 604, 468 (2022).

2) The manuscript is written in a very dense way, with little attention to the general readership. This can be improved in major ways. For example, the authors concluded the presence of FM-Z order, without any effort to define it clearly. The same goes with XY-order. Of course, to a few experts, this may seem obvious. But the authors are not submitting the paper to, say, J. Mag. & Mag. Mat. The Hamiltonian is in the SI. If the authors do not want to move it to the introduction section, they can at least briefly educate the readers early on.

3) Interlayer excitons are not pure bosons, but also possess major fermionic characters due to the charge separation. In fact, the authors' prior paper (ref 21) clearly showed how interlayer excitons and doped electrons can have very similar functions in forming correlated insulators. In this regard, there are a large number of papers on spin ordering in electron/hole doped moire systems. The connection of the authors findings to this body of literature would be desirable.

4) In ref. 21, the authors established the correlated insulator state from both doped electrons and photo-created interlayer excitons. They authors clearly can control the doping levels. How does the exciton spin order depend on doping?

5) In the figures on helicity, the authors repeated the date a few times. I can not tell if they are just repeated in the plots. Please clarify.

REVIEWER COMMENTS

Reviewer #1 (Remarks to the Author):

“Tunable exciton valley pseudospin orders in moiré superlattices” by Xiong et. al. explore the low temperature excitonic properties of a R stacked WSe₂/WS₂ vertical heterostructure. The paper using both pump-probe and pump only PL measurements, focuses on the exploration of excitonic properties link to the 0-degree alignment resulting moiré superlattice. The authors observe that close to filling factor 1 a (xy) valley pseudospin order appears. They demonstrate that an increase of either the filling factor (from 1.1 to 1.25) or magnetic field (20mT) transition the system into a Ferromagnetic Z phase. One may regret that no other type of layer alignment (H stacking) or other filling factor (-1.1 and -1.25 for example) are explored, as it could have given more ground and context to the paper main findings. Nevertheless, the findings of the paper are new and are presented in a clear manner. The resulting interpretation in term of spin $\frac{1}{2}$ Bose Higgs model is convincing and I think that the results clear signature of Bose Einstein physics in excitons would deserve publication in Nature Communication. Nevertheless, there is in my opinion a major problem in the way the paper is presented:

We thank the reviewer for the positive assessment of our work and for the detailed comments that have helped to improve our manuscript.

- As stated in the introduction, the paper treats of exciton “magnetic” phase with “spin” order. But the spin is in fact in this case the pseudo spin the valley polarization (K and K’). I understand perfectly that spin and pseudo spin are just quantum numbers and are formally in great part inter-changeable. Nevertheless, this start bothering me as physical effect such as Zeeman, Ferromagnetism, out of plan spin polarization, and the resulting effect of external magnetic field on these phases start to be treated. Those are strongly related to spin (real spin) and concepts that would be trivial when applied to spin are not so when talking about the valley polarization (pseudo spin). I do not think that the findings of the paper or the interpretation are not correct, I just find that this could not only be treated/discussed only through the spin analogy without discussing further the implications more clearly. The paper would gain in quality by being more pedagogic and I would suggest to discuss more thoroughly the results in terms specific to valley polarization, with more identifiable bibliography, equations and schematic representations of the different valley polarized state describe in the paper.

Reply R1: We thank the reviewer for the helpful suggestions to improve the manuscript and have followed them to add explicit discussions on “spin” orders and physical effects in the context of valley. We have also included corresponding references related to valley pseudospin.

In particular, we have added the following discussions on valley pseudospin orders:

“... Spin-up and spin-down excitons correspond to (and hereafter refer to) excitons in the K and K' valleys – two degenerate but inequivalent corners of the hexagonal Brillouin zone – and are related by time-reversal symmetry¹. Similar to magnetism from real spins, exchange interaction between excitons of different valley pseudospins can lead to spontaneous ordering in the valley degree of freedom. For example, an out-of-plane ferromagnetic (FM- z) order polarizes spins to the same out-of-plane direction, which, in the exciton context, corresponds to a state where excitons are spontaneously polarized to the same valley. On the other hand, an in-plane spin is a coherent superposition of spin-up and spin-down. Similarly, excitons in an in-plane (xy) order are each a superposition state between the two valleys of equal amplitude^{2,3}.”

We have also added discussions on the effects of magnetic field in the valley context:

“... The most well-known effect from an out-of-plane magnetic field on spin is the Zeeman splitting that lifts the degeneracy between the two spins. Similarly, the dominant effect of magnetic field on excitons is an energy splitting between the two valleys, termed “valley Zeeman effect”^{4,5}, with a g -factor of 4 in monolayer TMD.”

Some other smaller concerns:

1- The paper introduction heavily focused on the description of pump probe experiment but the rest of the paper is in fact more focused on the pump only PL experiment, (3 figures versus 1).

Reply R2: We thank the reviewer for the helpful suggestions to elucidate the role of the pump probe measurements. Besides direct access to the “spin” exchange interaction (original Fig. 1), the pump probe measurements first allow us to unambiguously establish the system as a spin-1/2 Bose Hubbard model and determine the inter- and intra-species on-site interaction, which lays the foundation for further investigation of “spin” orders and their theoretical understanding. These data and discussions were mainly included in the methods section of the original manuscript. Based on suggestions of the reviewers, we have moved these discussions to the main text in the new section “Spin-1/2 Bose-Hubbard model”. The revised manuscript has two figures from the pump probe experiments (Fig. 1 and Fig. 2).

2- There is a fair amount of information pushed to the Method section at the end of the paper, with 9 different. The paper is made harder to read by the fact that it is not always clear when referred to Method in the paper which part of the Method is concern. I would suggest that some important discussion (From Discussion on the xy order and Theoretical phase diagram) to be transferred into the main text or at the minimum that the part of the method concern to be referred more clearly.

Reply R3: As detailed in reply R2, we have moved the discussions on spin-1/2 Bose Hubbard model to the main text as a new section “Spin- $\frac{1}{2}$ Bose-Hubbard model”. Following the reviewer’s suggestions, we have also added specific section names when referring to Methods.

3- Line 184 A giant GH value of 2.3 is announced in Figure 3 (d).

Reply R4: We thank the reviewer for the careful reading. We have corrected this sentence in the revised draft.

Reviewer #2 (Remarks to the Author):

Xiong et al. report experimental studies of inter-layer excitons confined in the moiré potential of a WS₂/WSe₂ bilayer. The authors claim that they have engineered an xy ferromagnetic ordered phase, accessed by mapping the Heisenberg Hamiltonian with the two species Bose-Hubbard (BH) model. Unfortunately I find that the authors claim is not sustained by their experimental observations. As a result I regret not to recommend the work for publication, for the following detailed reasons.

We sincerely thank the reviewer for his/her constructive criticism, which has helped us to improve both the presentation and contents of our work in the revised manuscript. Below we have addressed all the concerns in a point-by-point manner.

In Fig.1 the authors claim that they measure the chemical potential as a function of the filling of the moiré lattice. However, I am doubtful since in the panel b the amplitude of the probe PL strongly varies with the strength of the pump filling the lattice. Why is that? Why does the probe intensity increases until $\nu=1$ and then suddenly drops? Naively I conclude that the authors also measure the PL radiated by excitons injected by the pump so that their technique is not a pump-probe.

Reply R5: After reading the reviewer's comments carefully, we realized that major confusions are likely arising from understanding of our pump probe measurement. We apologize for not providing sufficient details in the previous draft. As detailed below, the (unpolarized) pump probe measurement (original Fig. 1b) directly measures chemical potential of interlayer exciton⁶. Therefore, **the sudden disappearance of peak I at $\nu = 1$ and emergence of peak II are originating from and directly indicating a chemical potential jump, i.e., an incompressible state of interlayer exciton⁶.** On the other hand, if PL radiated by excitons injected by the pump were also measured, then peak I would not suddenly drop at $\nu_{\text{ex}} = 1$ (Ref. ⁷⁻⁹).

As detailed in Ref.⁶, our pump probe measurement is an optical analog of electrical capacitance measurements. The pump light tunes the background exciton density, and the weak probe light injects a small number of additional excitons, whose emissions are isolated through lock-in detection. This allows us to isolate the energy it takes to add/remove one (probe-induced) interlayer exciton from the system on top of a given (pump-controlled) background exciton density, which is by definition the exciton chemical potential. At $\nu_{\text{ex}} = 1$, the exciton chemical potential jumps up by V since adding another exciton above $\nu_{\text{ex}} = 1$ necessarily forms a doublon site and therefore requires additional energy cost of the on-site interaction. We thereby expect a sudden jump of the pump-probe PL from peak I to peak II, i.e., sudden disappearance of peak I and emergence of peak II at $\nu_{\text{ex}} = 1$ (Fig. R1b).

The sudden jump of probe-induced PL energy at $\nu_{\text{ex}} = 1$ (Fig. R1b) is therefore a unique result from our pump-probe measurement, highlighting its capability of measuring exciton compressibility. In contrast, conventional PL measurement, which measures emissions from the pump-injected excitons, will not show a sudden change at $\nu_{\text{ex}} = 1$ (Ref. ⁷⁻⁹) as it cannot isolate low-energy excitations of excitons. In that case, peak I will saturate at $\nu_{\text{ex}} > 1$ instead of suddenly drop because the singly-occupied sites will continue contributing to PL at $\nu_{\text{ex}} > 1$.

Fig. R1: Unpolarized pump-probe spectroscopy. **a**, Schematics of our pump-probe spectroscopy on a type-II hetero-bilayer WSe₂/WS₂. The background interlayer-exciton density (red) is controlled by the pump light and the charge density is kept at zero. The probe light injects an extra interlayer-exciton (blue), whose response is isolated through lock-in detection. **b**, Exciton-filling dependence of probe-induced PL spectrum using unpolarized pump and probe light. A sudden jump of exciton chemical potential at $\nu_{\text{ex}}=1$ is observed (white arrow), indicating an incompressible state of exciton. Low (high) energy emission peak is denoted as peak I (II).

As a result, the incompressible nature of the interlayer exciton fluid is not transparent which is worrying since mapping of the xy-model with a 2-species BH system requires experimental conditions deep in the Mott insulating regime, e.g. with exactly one particle per site. At $\nu=1.3$ for example I do not find reasons to consider that excitons realise an insulating phase.

Reply R6: As detailed in Reply R5, our (unpolarized) pump probe measurements provide direct evidence of an incompressible state at $\nu_{\text{ex}} = 1$, indicating that the system is deep in the Mott insulating regime. As will be detailed in Reply R7, the polarization-resolved pump probe measurements further allow us to determine the on-site interactions and unambiguously establish the system as a spin-1/2 Bose Hubbard model.

The excitons are compressible at $\nu_{\text{ex}} > 1$ (e.g. $\nu_{\text{ex}} = 1.3$). However, magnet orders in Hubbard model or mapping into a t - J model¹⁰ does not require the system to be insulating or at exactly one particle per site. Instead, magnetic exchange interactions and orders are expected to emergence in a wide filling ranges of Hubbard model¹⁰⁻¹⁴ and have been experimentally demonstrated in both TMD moiré superlattices¹⁵⁻¹⁸ and cold atom system¹⁹⁻²¹. In fact, recent investigations of magnetism in Hubbard model have been focusing on away from one particle per site, such as Nagaoka magnetism in doped Mott insulator^{16,21-24}.

The title of Fig.1 is “spin dependent interactions”, however the authors do not report the spin-dependence of on-site interactions in Fig.1. As detailed in Ref.[37] this is essential. To possibly extract the phase diagram of the 2-species BH model, which maps into the Heisenberg model, the magnitude of the difference between intra- and inter-species interactions is critical. This difference is not quantified here and not explicitly considered. Mott-like insulators crucially depend on the difference between inter and intra-species on-site interactions. In the manuscript a single on-site interaction U is extracted unlike the 3 necessary scattering rates detailed in Ref.[37].

Reply R7: We agree with the reviewer that the intra- and inter-species interactions are critical to the phase diagram of a two-flavor BH model, and we apologize for not including detailed discussions in the main text. We have performed polarization-resolved pump probe measurements to directly determine the spin-dependent on-site interactions in our system and included a new section “Spin- $\frac{1}{2}$ Bose-Hubbard model” in the revised manuscript to establish the BH model before investigating spin orders in it. These new results and discussion unambiguously establish the system as a two-flavor BH model with $V = 30$ meV and $U \gg V$, where V and U are inter- and intra-species on-site interaction, respectively.

To extract spin-dependent on-site interaction, we use linearly polarized pump light to generate equal population of two “spins” in the background, an LCP probe light to selectively inject extra K valley (“spin”-up) excitons, and monitor “spin”-resolved responses by separately collecting RCP (K' valley) and LCP (K valley) PL induced by the probe light only (Fig. R2, a and b). The K and K' responses are rather similar at $\nu_{\text{ex}} < 1$ (Fig. R2c), which can be understood in the single exciton picture from a short valley lifetime that quickly relaxes valley polarization. We directly capture such relaxation process by time-resolved pump probe PL measurements (Fig. R2e). The pulsed probe light selectively injects K excitons at time zero, and the K' response remains unchanged. Afterwards the valley polarization quickly disappears over time, resulting in similar overall responses from the two valleys (Fig. R2c).

In contrast, the two valleys’ responses become dramatically different above $\nu_{\text{ex}} = 1$ (Fig. R2, d and f). Most strikingly, their responses have opposite signs for peak I. The negative K' response indicates that adding extra K excitons will decrease the number of

singly occupied K' sites. Such behavior is incompatible with the single exciton picture where adding K excitons always increase both K and K' exciton populations^{25,26} and is instead a unique consequence of exciton correlation. Our observation can be naturally understood from the Bose-Hubbard model: the K excitons injected by the probe form doublons at $\nu_{\text{ex}} > 1$, which will decrease the number of singly occupied sites by converting them into doublon sites. The decrease in K' sites therefore indicates that K excitons selectively form doublon sites with K' excitons (Fig. R3a). This is further confirmed by the perfect valley balance in doublon emission (peak II) regardless of experimental configuration (Fig. R3b), which requires K and K' excitons to be symmetric within any doublon.

Our results thus unambiguously establish a “spin”-dependent on-site repulsion between excitons. The ~ 30 meV jump of exciton chemical potential at $\nu_{\text{ex}} = 1$ (Fig. R1b) corresponds to the opposite-“spin” repulsion V , while the same-“spin” repulsion U is much greater than V . Consequently, doublons only form by two excitons of opposite “spins” like electrons in a Fermi-Hubbard model, which offers a realization of spin- $1/2$ Bose-Hubbard model.

We have included the above discussions in the revised manuscript.

Fig. R2: Spin-1/2 Bose-Hubbard model. **a,b,** Polarization-resolved probe-induced PL spectra as a function of exciton filling. A linear pump light is used to generate equal numbers of K and K' valley excitons in the background, while an LCP probe light selectively excites extra K valley excitons. K' valley (**a**) or K valley (**b**) PL response induced by the probe light is collected separately. UKK (UKK') refers to

pump injecting unpolarized excitons, probe injecting K valley excitons and PL detecting K (K') valley excitons. **c,d**, Linecuts of **(a)** and **(b)** at low **(c)** and high **(d)** exciton density. **e,f**, Probe-induced TRPL signals from peak I at low **(e)** and high **(f)** exciton density. A constant background signal from the CW linear pump light is subtracted. Pink boxes in **(c)** and **(d)** denote the spectral filter used to isolate peak I response. The negative signals in UKK' configuration **(f)** indicate that K excitons selectively form doublons with K' excitons.

Fig. R3: Response of a bosonic correlated insulator to transient extra K excitons. **a**, **(1)**, CW linear pump light injects equal number of K valley (red) and K' valley (grey) background excitons that form an exciton lattice. **(2)**, Pulsed circular polarized probe light transiently injects two extra K valley excitons, which takes two K' sites to form doublon sites (yellow circles). **(3)**, Doublon sites have equal probabilities to emit $K(K')$ excitons and leave a single site of $K'(K)$. **(4)**, After the doublons decay the system will have one more K single site and one less K' single site, giving rise to negative K' response and positive K response of peak I with equal amplitude. **b**, Probe-induced TRPL measurements of doublon emissions under linear pump and LCP probe configuration. The two valleys show identical amplitude and dynamics, consistent with the expectation of doublon emission.

In Fig.1c-g the authors do not justify why the increased probe intensity manifests ordering. For charges (electrons or holes) a similar effect has been reported and correlated to charge ordering through capacitance measurements. Here the same extrapolation can not be done. The sole pump-probe approach does not suffice to reach this conclusion since the authors do not extract the exciton compressibility.

Reply R8: We thank the reviewer for his/her comments. We realize that there are likely confusions in understanding our measurement scheme. As detailed in reply R5, the unpolarized pump probe measurement directly measures exciton compressibility

and indicates an incompressible state at $\nu_{\text{ex}} = 1$. In contrast, the polarization configuration in Fig. 1c-g of the original manuscript does not measure exciton compressibility, and instead directly probes “spin”-exchange interactions of excitons, as detailed below. Besides, incompressibility in a Hubbard model is not directly related to magnetic orders, as detailed in reply R6.

For convenience, we copy the original Fig. 1c-g here as Fig. R4a-f. We vary the pump light polarization and keep the probe LCP. Different pump polarization maintains background excitons of different spins, while the LCP probe light always injects spin-up excitons. The signal corresponds to probe-induced spin imbalance (the difference between K and K' emission induced by the probe light). Because the same probe light is used across Fig. R4a-c, any difference in probe-induced spin imbalance must originate from interaction with background excitons. This allows us to directly probe “spin”-exchange interactions of excitons by comparing Fig. R4a-c since they have background excitons of different spins, instead of from the increased probe intensity in a single panel.

For example, at low exciton density, Fig. R4a-c shows similar behaviors, indicating negligible spin exchange effects. At increasing exciton density, in contrast, the signals vary dramatically with pump polarization: the sharp signal enhancement at $\nu_{\text{ex}} \sim 1.1$ and quick drop at $\nu_{\text{ex}} > 1.2$ (black arrows) appear under both LCP and RCP pump (Fig. R4, a and b) but are absent in the linear pump case (Fig. R4c). This provides direct evidence of exciton “spin”-exchange interactions at $\nu_{\text{ex}} > 1$ and their rapid change with exciton filling.

Besides exciton “spin”-exchange interactions, the extreme sensitivity of the observe “spin” imbalance to external magnet field (Fig. R4d-f) suggests existence of “spin” order and adjacent transitions, which is further confirmed by pump-only measurements. We will discuss it in detail in Reply R10.

We would also like to clarify the connection between our measurement and electrical capacitance. Indeed, our pump probe measurement under unpolarized configuration (Fig. R1b) provides similar information as capacitance by isolating low energy “charge” excitations of the system. On the other hand, by controlling polarization configuration, our pump probe scheme further allows us to directly create and detect low energy “spin” excitations of the system, which is not possible in electrical capacitance. Consequently, electrical capacitance provides direct evidence of charge orders^{27,28} but not spin orders since compressibility does not directly relate to the latter. While our pump probe technique can probe both types of orders directly through the corresponding polarization configurations.

Fig. R4: Inter-site spin-dependent exciton interaction. **a-c**, Probe-induced spin imbalance as a function of background exciton fillings for LCP (**a**), RCP (**b**) and linear (**c**) pump respectively. For both LCP and RCP pump, a sharp enhancement of signals at $\nu_{\text{ex}} \sim 1.1$ followed by a quick drop at $\nu_{\text{ex}} \sim 1.2$ are observed (black arrows). In contrast, these features are missing under linear pump, indicating rapidly changing exciton spin interaction when doping slight away from the correlated insulator state. **d-f**, Evolution of probe-induced spin imbalance at $\nu_{\text{ex}} \sim 1.1$ under out-of-plane magnetic field B_z for LCP (**d**), RCP (**e**) and linear (**f**) pump, respectively. The signals show sensitive and symmetric change under a small $|B_z| \sim 5$ mT. All the measurements are performed at base temperature 3 K unless specified.

For optically active excitons the compressibility is possibly deduced by monitoring PL intensity fluctuations with few sites resolution, see for instance Nature 609 485 (2022). Following the above paragraph I hardly see how the authors can justify that excitons realise any incompressible phase.

Reply R9: As detailed in Reply R5 and Ref.⁶, the unpolarized pump probe measurement directly measures exciton compressibility. On the other hand, compressibility is not directly related to the “spin” orders of excitons (see Reply R6), which is the focus of this study. Instead, we directly access exciton “spin” physics through polarization-controlled pump probe measurements, as detailed in Reply R8.

In Fig.2 the authors study the PL polarisation as a function of the pump polarisation (with no probe). Fig.2.b shows that the PL has a polarisation that increasingly follows the one from the pump while the moiré filling is increased. At ν around 1.3 Fig.2.b shows that the PL is well polarised by the pump (see Fig.S9) when no magnetic field is

applied. Then the authors compute a so-called generalised ellipticity (GH). However I find that GH does not clarify pseudo-spin ordering. I understand that the authors have tried to rely on the best possible calibration, but GH is computed by dividing the PL ellipticity by the one of the pump which diverges for linear pump polarisation. I am then not comfortable with the authors analysis that uses GH as a sort of order parameter. This is not suitable and in fact to my view GH artificially exacerbates differences between pump and PL polarisations. As a matter of fact, in Fig.2.b the polarisation is almost unchanged between $\nu=1.1$ and 1.3 whereas Fig.2.d reports larger differences.

Reply R10: We thank the reviewer for the important question about GH. Below we would like to first argue that GH provides direct information of exciton “spin” orders; and then explain the reliability of our GH calculation.

The GH is defined as the ratio between the PL and pump helicity, $GH = \eta_{PL}/\eta_{pump}$. In the case of no spin order, the LCP and RCP components of pump light independently contribute to PL emission. Therefore, the PL helicity should be proportional to the pump helicity, giving rise to a constant ratio, i.e., an GH independent on the pump helicity. In contrast, an exciton spin order generates a mean field that depends on all exciton spins in the system and thus on the pump polarization. PL helicity would therefore deviate from linear relation with pump helicity, leading to a GH changing with the pump helicity. In this sense, the GH isolates effects from exciton “spin” orders from single-exciton effects, thereby providing direct evidence of exciton “spin” orders.

We provide two concrete examples to further illustrate the physical meaning of GH. As the first example, Fig. R5a and b compares the PL raw helicity and GH for two distinctively different measurement configurations: $\nu_{ex}=0.01$ at $T = 3$ K (blue color) and $\nu_{ex} = 1.3$ at $T = 60$ K (red color). The PL raw helicities are significantly different between the two cases. However, since such difference is due to single particle effects and neither has exciton “spin” orders, the GH in both cases is a constant. As a contrasting example, Fig. R5c and d compare the PL raw helicity and GH for $\nu_{ex}=1.1$ (magenta color) and 1.39 (orange color) at 3 K, the cases mentioned in the comments of the reviewer. The raw helicity already shows large differences, and the nontrivial shapes of GH further indicate the existence of exciton “spin” orders. **These examples demonstrate that the GH does not exaggerate the difference in PL raw helicity, and instead isolates the contribution from exciton “spin” orders.** This can also be directly seen from the definition of GH: because the same η_{pump} is used to calculate GH for all exciton fillings, no artefact or exaggeration will be introduced.

Now we explain the reliability of GH calculation. As the reviewer pointed out, the calculation of GH requires careful treatment at near linear pump polarization since both the numerator and denominator approach zero in η_{PL}/η_{pump} . Nevertheless, we have carefully considered and resolved this issue in the manuscript. First, **all observations and conclusions still hold even without the data at small η_{pump} .** As discussed above and in the manuscript, the “V” shape GH that exceeds unity is smoking gun evidence of an

FM-z order of excitons at $\nu_{\text{ex}}=1.3$. Fig. R6 shows a close-up of the GH in this case. GH exhibits non-trivial shapes and exceeds unity over a large range of η_{pump} well beyond those near linear polarization (green shaded area). For example, if we only considered the data at $|\eta_{\text{pump}}|>0.5$, GH still exhibits a “V” shape that decreases from $\text{GH}=1.25>1$ at $|\eta_{\text{pump}}|=0.5$ to $\text{GH}=0.85$ at $|\eta_{\text{pump}}|=1$. The conclusion of FM-z order would therefore still hold.

We further argue that **GH at small η_{pump} can also be reliably obtained after careful consideration**. Although both η_{PL} and η_{pump} approaches zero, their ratio is still well-defined since in this case $\text{GH} = \eta_{\text{PL}}/\eta_{\text{pump}} = d\eta_{\text{PL}}/d\eta_{\text{pump}}$, or simply the slope in the $\eta_{\text{PL}}-\eta_{\text{pump}}$ plot (Fig. R5c). Such slope can be directly and reliably computed without involving any diverging term, which we used to obtain GH at small η_{pump} (within the green shaded region in Fig. R6). The fact that GH connects smoothly at small η_{pump} directly confirms its reliability.

Fig. R5: Evidence of exciton spin orders from generalized helicity (GH). **a,b**, PL raw helicity η_{PL} (**a**) and GH (**b**) at $\nu_{\text{ex}}=0.01$, $T = 3$ K (blue color) and at $\nu_{\text{ex}}=1.3$, $T = 60$ K (red color). GH is a constant in both cases. **c,d**, PL raw helicity η_{PL} (**c**) and GH (**d**) for $\nu_{\text{ex}}=1.1$ (magenta color) and 1.3 (orange color) at 3 K. The nontrivial shapes of GH indicate the existence of exciton “spin” orders.

Fig. R6. Generalized helicity (GH) at $\nu_{\text{ex}}=1.39$ and zero magnetic field as a function of pump helicity. Regions near linear pump ($|\eta_{\text{pump}}|<0.05$) are labelled by green shades.

In general I find that the experiments reported in Fig.2 and Fig.3 only reveal an optically induced magnetisation, but I find no evidence for either spin ordering or incompressible exciton phases. For example, at ν around 1.3 the moire lattice has singly and doubly filled sites. Ref.[37] shows that theoretically this strongly modifies the structure of insulating Mott-like phases. In the manuscript the authors do not address this aspect and the experimental results vary weakly above $\nu=1$ which is very surprising.

Reply R11: Our evidence of spin orders and the relevance of compressibility has been discussed in detail in Reply R6 to R10. Therefore, here we will mainly discuss the filling dependence at $\nu_{\text{ex}}>1$. We agree with the reviewer that adding particles to the Mott-like state should strongly modify the “spin” exchange interaction and orders, and our experimental results directly confirm such prediction. In fact, this is the main conclusion of our manuscript as the title and abstract explicitly states “tunable exciton valley-pseudospin orders” from “exciton density and magnetic field”.

We have also thoroughly and explicitly discussed the exciton density dependence throughout the manuscript. For example, we emphasized the contrasting behaviors at $\nu_{\text{ex}}=1.1$ and $\nu_{\text{ex}}=1.3$ in the original Fig. 1; the distinctively different shapes of GH and their magnetic field dependence in original Fig. 2 and 3; and later attribute them to distinct exciton “spin” orders. The experimental phase diagram (original Fig. 4c) also provides a clear demonstration of how exciton filling dramatically modifies exciton “spin” orders at $\nu_{\text{ex}}>1$, where the order parameter changes sign (color changes from blue to red) around $\nu_{\text{ex}}=1.25$. In the theory section, we have also explicitly discussed the expected behaviors of a spin-1/2 BH model when doped above $\nu_{\text{ex}}=1$: extra particles should lead to increasing FM-z exchange interaction from Nagaoka-type mechanism and therefore a transition from an FM-xy to FM-z order.

In Fig.4 the authors tentatively build a phase diagram based on GH and argue that pseudo-spin ordering persists up to around 30-50K. I regret but I find this conclusion is

in contradiction with other measurements of the manuscript. Precisely, Eq.(3) in Ref.[37] shows that xy-order results from the competition between J_{\perp} and h . Here, the authors explore a regime where U is around V_a and V_b , and t_a is around t_b , so that J_z is vanishing and h reduces to h_{ext} . xy-ordering theoretically melts when kT is around J_{\perp} , so here around 30-50 K, i.e. 3-5 meV. At the same time, xy ordering melts when h becomes greater than J_{\perp} and the authors find that this occurs at 5-10 mT (Fig.1.h). This comparison leaves orders of magnitude difference, revealing that J_{\perp} is not around 5 meV despite the authors claim. In fact, I am not surprised here since a simple calculation shows that the tunnelling coefficient (t_a and t_b) are of the order of 0.05 meV for a moiré depth estimated around 100 meV. Since the authors measure that U is around 30 meV, J_{\perp} is reduced to around 1 micro-eV so that xy-ordered phases are far out of experimental reach here. The linewidth of the PL (10-20 meV) further supports my conclusion.

Reply R12: We thank the reviewer for the careful assessment. However, we would like to point out that the numbers and equations used by the reviewer are not entirely accurate. Once corrected, all observations here are consistent.

First, we show that the J_{xy} estimated from temperature dependence is consistent with the expectation from the BH model. The xy order in a triangular lattice is expected to melt at $k_B T_c = 6J_{xy}$. Using the measured $T_c = 35$ K, we estimate a $J_{xy} \sim 0.5$ meV. Meanwhile, we can also independently estimate J_{xy} from $J_{xy} = 4t^2/V$, where t and V are intra (inter)-species hopping and inter-species on-site repulsion, respectively (Ref. ²⁹, note that our notation of U and V is different from the reference). As detailed in Reply R7, our measurements allow us to directly determine $V = 30$ meV. Using the typical values of experimentally obtained hopping energy $t=1\sim 5$ meV (Ref. ^{15,30,31}), we estimate $J_{xy} = 0.1\sim 3$ meV, which is consistent with the temperature dependence.

Second, we argue that J_z is not vanishing, and instead playing a critical role. As detailed in Reply R7, our measurements indicate that U is much greater than V , instead of similar to V . Consequently, J_z will not vanish at $\nu_{\text{ex}} = 1$. On the other hand, $U > V$ ensures that $|J_z| < |J_{xy}|$ at $\nu_{\text{ex}} = 1$. The BH model therefore predicts an FM-xy order around $\nu_{\text{ex}}=1$. Further increasing ν_{ex} is expected to strongly modify the system, as the reviewer mentioned. In particular, J_z is expected to increase with ν_{ex} through Nagaoka-type ferromagnetism^{10,12,32}, which eventually makes $|J_z| > |J_{xy}|$ and leads to a transition between FM-xy to FM-z order. The entire phase diagram is therefore governed by a competition between J_{xy} and J_z .

Last, we show that the observed sensitive magnetic field dependence is incompatible with a small J_{xy} , the scenario suggested by the reviewer, and instead confirms the above picture of competing orders. In the case of a small J_{xy} , a small external magnetic field h_{ext} would indeed destroy the xy order and align spins to out-of-plane. However, changing the sign of h_{ext} would also reverse the sign of spin polarization. In contrast, experimentally we observe symmetric field dependence in all

measurements, thereby directly excluding a simple field-induced polarization. In addition, following the reviewer's estimation, the observed saturation field of ~ 10 mT (Fig. R4d-f) corresponds to an energy scale much smaller than the measurement temperature, which should not happen in non-interacting spin systems with weak exchange interaction. On the other hand, the sensitive magnetic field dependence can be naturally accounted for by the existence of competing orders, since a weak h_{ext} comparable to $|J_{xy} - J_z|$ is sufficient to suppress the xy order and promote the FM-z order.

To conclude, Xiong et al. claim that interlayer excitons realise xy -ordered phases in a moiré lattice. This is a very strong claim and to my view the experimental results do not provide a ground to reach such conclusion. I only deduce from their work that they report optically induced magnetism which does not deserve publication in Nat. Comm.

Reply R13. We hope that the added results and discussions have clarified the reviewer's concerns. In summary, our study reveals both an xy -ordered phase of excitons and its transition into an FM-z order through tuning either exciton filling or external magnetic fields. These results pave the way for discovering other exotic bosonic phases in solid state systems and lay the foundation for novel nano-photonics devices based on correlated excitons.

Reviewer #3 (Remarks to the Author):

This manuscript reports on the finding of magnetic orders of excitons in the angle-aligned moire superlattice of WSe₂/WS₂. This is a very interesting follow up of the discovery of exciton insulator in this system by the same group (ref. 21). Using a strong CW pump and weak pulsed probe, the author showed convincing evidences for magnetic orders from the valley pseudo spins, as experimentally revealed by circular polarized PL, of the photo-excited interlayer excitons. The sensitivity of the observed spin orders to very weak external magnetic field along the z-direction suggests the presence of adjacent phase transitions. While these findings are of interest to the broad community working on moire physics, the writing needs to be substantially improved before considerations for publication. Here are issues that require attention.

We thank the reviewer for recognizing the significance of our work and helpful suggestions to improve its quality.

1) The (valley) spin order from interlayer excitons can be considered as light-induced magnetism. It is strange that the authors failed to discuss relevant publications on the topic, e.g., the paper of Xiaodong Xu and coworkers Nature 604, 468 (2022).

Reply R13: We thank the reviewer for pointing out related literature. Indeed, our study fit broadly into the concept of light-induced magnetism. We have followed the reviewer's suggestions and added discussions and references within this context to the introduction of the revised manuscript. On the other hand, the "magnetism" observed here is from spontaneously ordering of excitons, while in all previous studies magnetism is from ordering of free electrons (or holes). Therefore, the physics, implications and experimental approaches are fundamentally different, as detailed below.

Xu. et al's work focuses on the behavior of free carriers under the influence of light-induced excitons. The reported magnetism originates from free carriers instead of excitons, though the electron-electron interaction can be modified by the presence of excitons. In contrast, our study investigates exciton-exciton interactions and the resultant exciton pseudospin orders, while the **free carrier is kept at zero density throughout the study**. In fact, adding free carriers to the system will destroy the exciton "spin" orders (see Reply R16), which directly excludes the contribution of free carriers to the observed exciton "spin" orders. In addition, the observed exciton xy order and its filling- and magnetic-field-tuned transitions are both unique consequences from the bosonic statistics and are distinctively different from fermionic electrons (see Reply R15). Last but not least, the exciton density in Xu's study is estimated to be around 10^{11} cm⁻², which is still in the dilute regime compared to the moiré density of $\sim 2 \times 10^{12}$ cm⁻² in WSe₂/WS₂. In contrast, our study focuses on the regime of large exciton density $\nu_{ex} \sim 1$, which is required for exciton spin orders.

2) The manuscript is written in a very dense way, with little attention to the general readership. This can be improved in major ways. For example, the authors concluded the presence of FM-Z order, without any effort to define it clearly. The same goes with XY-order. Of course, to a few experts, this may seem obvious. But the authors are not submitting the paper to, say, J. Mag. & Mag. Mat. The Hamiltonian is in the SI. If the authors do not want to move it to the introduction section, they can at least briefly educate the readers early on.

Reply R14: We thank the reviewer for the helpful comments to improve our manuscript. We have followed the suggestions and included detailed discussions of the relevant concept and models at the beginning, as detailed below.

First, we have added explicit explanations of the various “spin” orders, especially in the context of valley pseudospin:

“... Spin-up and spin-down excitons correspond to (and hereafter refer to) excitons in the K and K' valleys – two degenerate but inequivalent corners of the hexagonal Brillouin zone – and are related by time-reversal symmetry¹. Similar to magnetism from real spins, exchange interaction between excitons of different valley pseudospins can lead to spontaneous ordering in the valley degree of freedom. For example, an out-of-plane ferromagnetic (FM- z) order polarizes spins to the same out-of-plane direction, which, in the exciton context, corresponds to a state where excitons are spontaneously polarized to the same valley. On the other hand, an in-plane spin is a coherent superposition of spin-up and spin-down. Therefore, excitons in an in-plane (xy) order are each a superposition between the two valleys of equal amplitude^{2,3}.”

Second, we have added a new section titled “**Spin-1/2 Bose-Hubbard model**” as the first section of the manuscript, where we explicitly discussed the relevant Hamiltonian as well as how we experimentally establish such Hamiltonian and the parameters within. For convenience, we have copied the new section and figure below.

“... We then switch experimental configurations to inspect spin excitations of excitons. The minimum model to account for exciton “spins” is a two-component Bose Hubbard model²⁹, given by

$$H = \sum_{\langle i,j \rangle, \alpha} -tb_{i,\alpha}^\dagger b_{j,\alpha} + h.c. + \sum_{i,\alpha} U(n_{i,\alpha} - 1/2)^2 + \sum_i V n_{i,1} n_{i,2}$$

Here the $\alpha=1,2$ label K and K' valley pseudo-spin of excitons, t is hopping between nearest neighboring sites, and the interactions between excitons consist of intra-species repulsion U and inter-species repulsion V . To establish such a model and separately determine U and V , we use linearly polarized pump light to generate equal population of two “spins” in the background, an LCP probe light to selectively inject extra K valley

(“spin”-up) excitons, and monitor “spin”-resolved responses by separately collecting RCP (K' valley) and LCP (K valley) PL from the probe light only (Fig. R7, c and d). The K' and K responses are rather similar at $\nu_{\text{ex}} < 1$ (Fig. R7e), which can be understood in the single exciton picture from a short valley lifetime that quickly relaxes valley polarization. We directly capture such relaxation process by time-resolved pump probe PL measurements (Fig. R7g). The pulsed probe light selectively injects K excitons at time zero, and the K' response remains unchanged. Afterwards the valley polarization quickly disappears over time, resulting in similar overall responses from the two valleys (Fig. R7e).

In contrast, the two valleys' responses become dramatically different above $\nu_{\text{ex}} = 1$ (Fig. R7f). Most strikingly, their responses have opposite signs for peak I. The negative K' response indicates that adding extra K excitons will decrease the number of singly occupied K' sites. Such behavior is incompatible with the single exciton picture where adding K excitons always increase both K and K' exciton populations^{25,26} and is instead a unique consequence of exciton correlation. Our observation can be naturally understood from the Bose Hubbard model: the K excitons injected by the probe form doublons at $\nu_{\text{ex}} > 1$, which will decrease the number of singly occupied sites by converting them into doublon sites. The decrease in K' sites therefore indicates that K excitons selectively form doublon sites with K' excitons (Extended Data Fig. 2a). This is further confirmed by the perfect valley balance in doublon emission (peak II) regardless of experimental configuration (Extended Data Fig. 2b), which requires K and K' excitons to be symmetric within any doublons.

Our results thus unambiguously establish a “spin”-dependent on-site repulsion between excitons. The ~ 30 meV jump of exciton chemical potential at $\nu_{\text{ex}} = 1$ (Fig. R7b) corresponds to the opposite-“spin” repulsion V , while the same-“spin” repulsion U is much greater than V . Consequently, doublons only form by two excitons of opposite “spins” like electrons in a Fermi-Hubbard model, which offers a rare realization of spin- $\frac{1}{2}$ Bose-Hubbard model....”

Fig. R7: Spin-1/2 Bose-Hubbard model. **a**, Schematics of our pump-probe spectroscopy on a type-II hetero-bilayer WSe₂/WS₂. The background interlayer-exciton density (red) is controlled by the pump light and the charge density is kept at zero. The probe light injects an extra interlayer-exciton (blue), whose response is isolated through lock-in detection. **b**, Exciton-filling dependence of probe-induced PL spectrum using unpolarized pump and probe light. A sudden jump of exciton chemical potential at $\nu_{\text{ex}}=1$ is observed (white arrow), indicating an incompressible state of exciton. Low (high) energy emission peak is denoted as peak I (II). **c,d**, Polarization-resolved probe-induced PL spectra as a function of exciton filling. A linear pump light is used to generate equal numbers of K and K' valley excitons in the background, while an LCP probe light selectively excites extra K valley excitons. K' valley (**c**) or K valley (**d**) PL response induced by the probe light is collected separately. UKK (UKK') refers to pump injecting unpolarized excitons, probe injecting K valley excitons and PL detecting K (K') valley excitons. **e,f**, Linecuts of (**c**) and (**d**) at low (**e**) and high (**f**) exciton density. **g,h**, Probe-induced TRPL signals from peak I at low (**g**) and high (**h**) exciton density. A constant background signal from the CW linear pump light is subtracted. Pink boxes in (**e**) and (**f**) denote the spectral filter used to isolate peak I response. The negative signals in UKK' configuration (**h**) indicate that K excitons selectively form doublons with K' excitons, showing the intra-species on-site interaction is much larger than the intra-species on-site interaction.

3) Interlayer excitons are not pure bosons, but also possess major fermionic characters due to the charge separation. In fact, the authors' prior paper (ref 21) clearly showed how interlayer excitons and doped electrons can have very similar functions in forming correlated insulators. In this regard, there are a large number of papers on spin ordering in electron/hole doped moire systems. The connection of the authors findings to this body of literature would be desirable.

Reply R15: We thank the reviewer for the helpful suggestions to discuss comparison between exciton “spin” orders observed here and electron magnetism reported in literatures. Below we first provide a brief overview of spin ordering of electrons, and then show that the exciton “spin” order here is distinctively different.

As the reviewer points out, spin orderings of electrons have been widely observed in both TMD^{15,16,33–36} and graphene moiré systems^{37,38}. They can be divided into two classes:

1. The spin ordering is associated with band topology. These systems involve two or more orbitals, and the magnetism is intimately connected to topological states such as Chern insulator. Example systems include magic-angle twist bilayer graphene³⁷, ABC trilayer graphene/hBN³⁸, AB stacked MoTe₂/WSe₂ heterobilayers³³ and AA stacked MoTe₂ homobilayers^{34,36}.

2. The underlying band is non-topological. These systems can be described by a single band Fermi Hubbard model and the spin exchange/orders can be understood from doping a Mott insulator. Examples include frustrated magnetic interaction in WSe₂/WS₂^{15,35} and kinetic magnetism in MoSe₂/WS₂¹⁶.

The exciton spin orders reported here belong to a fundamentally new category with distinctive different physics. Since the single-particle exciton bands are non-topological, the physics here can be captured by a Hubbard model. Nevertheless, the bosonic nature of exciton gives rise to distinct behaviors as compared to the Fermion case. In particular, a Bose Hubbard model predicts in-plane FM super-exchange interaction at $\nu = 1$ ²⁹ and out-of-plane FM interaction from Nagaoka-type mechanism at $\nu > 1$; while a Fermi Hubbard model predicts antiferromagnetic (AFM) super-exchange interaction along all directions at $\nu = 1$ ^{39,40}, and the Nagaoka FM is also isotropic instead of favoring the z direction^{32,39}. Our experimental observation of transition between an xy order to an FM- z order is therefore a unique consequence of Bose-Einstein statistics.

We would also like to comment on the relevance of the exciton internal structure. Indeed, excitons are not a simple boson due to the separation between the electron and hole within. However, the binding energy of interlayer exciton (>100 meV)⁴¹ is much larger than all other relevant energy scales such as on-site repulsion, hopping and exchange interaction. Equivalently, the Bohr radius of exciton is much smaller than the relevant length scale such as the moiré period. Consequently, it is typically a good approximation to treat exciton as a single boson instead of two separate fermions^{42–44}, as confirmed by our experimental observation.

We have included the above discussions in moiré systems in the revised methods section.

4) In ref. 21, the authors established the correlated insulator state from both doped

electrons and photo-created interlayer excitons. They authors clearly can control the doping levels. How does the exciton spin order depend on doping?

Reply R16: Experimentally, we found that adding free carriers to the system strongly suppressed the exciton spin orders. In Fig. R8b we show the B-field dependence of GH at exciton filling $\nu_{ex}\sim 1.1$ and electron filling $\nu_e\sim 0.2$. GH is largely a constant over the pump helicity and does not show sensitive B-field transition dependence. This indicates much weaker exciton “spin” exchange interaction and orders upon adding free carriers. Theoretically, considering both excitons and free electrons requires a multi-species, multi-flavor model with exponentially more complex analysis, which is beyond the scope of this study. We therefore did not include the results in the original manuscript.

Fig. R8: Magnetic field dependence of PL raw helicity η_{PL} (a) and GH (b) at $\nu_{ex}=1.1$ and $\nu_e=0.2$ from 0 to 40 mT. GH keeps largely a constant and no sensitive magnetic field transition is observed.

5) In the figures on helicity, the authors repeated the data a few times. I can not tell if they are just repeated in the plots. Please clarify.

Reply R17: We apologize for the confusion. To isolate the effects from exciton “spin” orders in PL helicity, we have presented the data in three different ways: 1) PL raw helicity η_{PL} , 2) generalized helicity $GH = \eta_{PL}/\eta_{pump}$, and 3) normalized GH (normalized to the value at $\eta_{pump}=1$). For example, Fig.3 b-d in the revised draft are different analyses and presentations from the same data source: η_{PL} (b), GH (c) and normalized GH (d), respectively. The exact quantity presented is also provided in the vertical axis titles and figure captions.

Reference:

1. Xu, X., Yao, W., Xiao, D. & Heinz, T. F. Spin and pseudospins in layered transition metal dichalcogenides. *Nat Phys* **10**, 343–350 (2014).
2. Kim, J. *et al.* Ultrafast generation of pseudo-magnetic field for valley excitons in WSe₂ monolayers. *Science (1979)* **346**, 1205–1208 (2014).
3. Ye, Z., Sun, D. & Heinz, T. F. Optical manipulation of valley pseudospin. *Nat Phys* **13**, 26–29 (2017).
4. Srivastava, A. *et al.* Valley Zeeman effect in elementary optical excitations of monolayer WSe₂. *Nat Phys* **11**, 141–147 (2015).
5. Aivazian, G. *et al.* Magnetic control of valley pseudospin in monolayer WSe₂. *Nat Phys* **11**, 148–152 (2015).
6. Xiong, R. *et al.* Correlated insulator of excitons in WSe₂/WS₂ moiré superlattices. *Science (1979)* **380**, 860–864 (2023).
7. Park, H. *et al.* Dipole ladders with large Hubbard interaction in a moiré exciton lattice. *Nat Phys* **19**, 1286–1292 (2023).
8. Lian, Z. *et al.* Valley-polarized excitonic Mott insulator in WS₂/WSe₂ moiré superlattice. *Nat Phys* **20**, 34–39 (2024).
9. Gao, B. *et al.* Excitonic Mott Insulator in a Bose-Fermi-Hubbard System of Moiré WS₂/WSe₂ *Heterobilayer*. arxiv: 2304.09731v1 (2023)
10. Seifert, U. F. P. & Balents, L. Spin Polarons and Ferromagnetism in Doped Dilute Moiré-Mott Insulators. *Phys Rev Lett* **132**, 046501 (2024).
11. Morera, I. *et al.* High-temperature kinetic magnetism in triangular lattices. *Phys Rev Res* **5**, L022048 (2023).
12. Davydova, M., Zhang, Y. & Fu, L. Itinerant spin polaron and metallic ferromagnetism in semiconductor moiré superlattices. *Phys Rev B* **107**, (2023).
13. Morera, I. & Demler, E. *Itinerant Magnetism and Magnetic Polarons in the Triangular Lattice Hubbard Model*. arxiv: 2402.14074v1. (2024).
14. Samajdar, R. & Bhatt, R. N. *Nagaoka Ferromagnetism in Doped Hubbard Models in Optical Lattices*. arxiv: 2305.05683 (2023).
15. Tang, Y. *et al.* Simulation of Hubbard model physics in WSe₂/WS₂ moiré superlattices. *Nature* **579**, 353–358 (2020).
16. Ciorciaro, L. *et al.* Kinetic magnetism in triangular moiré materials. *Nature* **623**, 509–513 (2023).
17. Anderson, E. *et al.* Programming correlated magnetic states with gate-controlled moiré geometry. *Science (1979)* **381**, 325–330 (2023).
18. Zhao, W. *et al.* *Emergence of Ferromagnetism at the Onset of Moiré Kondo Breakdown*. arXiv:2310.06044 (2023).
19. Salomon, G. *et al.* Direct observation of incommensurate magnetism in Hubbard chains. *Nature* **565**, 56–60 (2019).
20. Koepsell, J. *et al.* Imaging magnetic polarons in the doped Fermi–Hubbard model. *Nature* **572**, 358–362 (2019).
21. Xu, M. *et al.* Frustration- and doping-induced magnetism in a Fermi–Hubbard simulator. *Nature* **620**, 971–976 (2023).

22. Prichard, M. L. *et al.* Directly imaging spin polarons in a kinetically frustrated Hubbard system. arXiv:2308.12951 (2023).
23. Lebrat, M. *et al.* Observation of Nagaoka Polarons in a Fermi-Hubbard Quantum Simulator. arXiv:2308.12269 (2023).
24. Tao, Z. *et al.* Observation of spin polarons in a frustrated moiré Hubbard system. arXiv:2307.12205 (2023).
25. Jin, C. *et al.* Ultrafast dynamics in van der Waals heterostructures. *Nat Nanotechnol* **13**, 994–1003 (2018).
26. Jiang, Y., Chen, S., Zheng, W., Zheng, B. & Pan, A. Interlayer exciton formation, relaxation, and transport in TMD van der Waals heterostructures. *Light Sci Appl* **10**, 72 (2021).
27. Li, T. *et al.* Charge-order-enhanced capacitance in semiconductor moiré superlattices. *Nat Nanotechnol* **16**, 1068–1072 (2021).
28. Tomarken, S. L. *et al.* Electronic Compressibility of Magic-Angle Graphene Superlattices. *Phys Rev Lett* **123**, 046601 (2019).
29. Altman, E., Hofstetter, W., Demler, E. & Lukin, M. D. Phase diagram of two-component bosons on an optical lattice. *New J Phys* **5**, 113–113 (2003).
30. Mak, K. F. & Shan, J. Semiconductor moiré materials. *Nat Nanotechnol* **17**, 686–695 (2022).
31. Li, T. *et al.* Continuous Mott transition in semiconductor moiré superlattices. *Nature* **597**, 350–354 (2021).
32. Nagaoka, Y. Ferromagnetism in a Narrow, Almost Half-Filled s Band. *Physical Review* **147**, 392–405 (1966).
33. Li, T. *et al.* Quantum anomalous Hall effect from intertwined moiré bands. *Nature* **600**, 641–646 (2021).
34. Park, H. *et al.* Observation of fractionally quantized anomalous Hall effect. *Nature* **622**, 74–79 (2023).
35. Tang, Y. *et al.* Evidence of frustrated magnetic interactions in a Wigner–Mott insulator. *Nat Nanotechnol* **18**, 233–237 (2023).
36. Xu, F. *et al.* Observation of Integer and Fractional Quantum Anomalous Hall Effects in Twisted Bilayer MoTe₂. *Phys Rev X* **13**, (2023).
37. Serlin, M. *et al.* Intrinsic quantized anomalous Hall effect in a moiré heterostructure. *Science (1979)* **367**, 900–903 (2020).
38. Chen, G. *et al.* Tunable correlated Chern insulator and ferromagnetism in a moiré superlattice. *Nature* **579**, 56–61 (2020).
39. Tasaki, H. The Hubbard model - an introduction and selected rigorous results. *Journal of Physics: Condensed Matter* **10**, 4353–4378 (1998).
40. Lee, P. A., Nagaosa, N. & Wen, X. G. Doping a Mott insulator: Physics of high-temperature superconductivity. *Rev Mod Phys* **78**, 17–85 (2006).
41. Regan, E. C. *et al.* Emerging exciton physics in transition metal dichalcogenide heterobilayers. *Nat Rev Mater* **7**, 778–795 (2022).
42. Wu, F., Lovorn, T. & MacDonald, A. H. Topological Exciton Bands in Moiré Heterojunctions. *Phys Rev Lett* **118**, 147401 (2017).
43. Götting, N., Lohof, F. & Gies, C. Moiré-Bose-Hubbard model for interlayer excitons in twisted transition metal dichalcogenide heterostructures. *Phys Rev B* **105**, 165419 (2022).
44. Yu, H., Liu, G.-B., Tang, J., Xu, X. & Yao, W. Moiré excitons: From programmable quantum

emitter arrays to spin-orbit-coupled artificial lattices. *Sci Adv* **3**, e1701696 (2017).

Reviewers' Comments:

Reviewer #1:

Remarks to the Author:

The authors have answered convincingly to all my previous comments, they have furthermore improved the way the paper is written making it clearer for the reader. As I had said in my previous comments, the presented results are quite new and interesting and the very good agreement between the experimental data and simulation is convincing. In conclusion I think the paper deserve publication in Nature Communication.

Reviewer #2:

Remarks to the Author:

I have carefully read the revised manuscript from Xiong et al. together with the resubmission letter.

After this rebuttal I recommend not to publish this manuscript, which makes claims unsupported by experimental observations. In fact, I dispute the authors hypothesis and interpretations. Compared to the state-of-the-art (Nature Physics 19, 1286 (2023)) I find that this manuscript does not provide novel physics, and instead blurs the clear understanding given in Nature Physics 19, 1286 (2023).

1) The authors assume that excitons realise a Mott insulator at unity filling. This hypothesis lays the ground for all the interpretation. However, it is not verified and I dispute its validity: probe excitons either fill empty moire sites, or induce doublons. In Fig.1, around an average unity filling probe excitons still populate empty sites, as shown by line I, which by definition implies that the pump beam has not induced a Mott insulator, otherwise all sites would be occupied by one exciton. Moreover, the pump probe method introduced here does not allow to extract the chemical potential. It only provides the magnitude of on-site interactions, as already done in the manuscript by the Xu group (Nature Physics 19, 1286 (2023)) without probe beam. In the conclusion of the paper from Park et al. it is clearly stated that compressibility measurements are critical, and I agree with this. Such measurements are essential, as shown for instance in Nature 609, 485 (2022). Measuring exciton compressibility in the moire potential marks the next milestone in the field to my point of view.

In my previous report I already stressed that the intensity difference in the probe induced PL (Fig.1.b) while varying the pump beam power shows that the experimental approach does not allow to fully isolate the influence of the probe. Unfortunately, the authors have not replied to this concern that was very important because to my view it underlines that here the compressibility is not extracted.

2) The authors present that increasing the moire filling enhances the optically induced exciton polarisation. However, this observation is not novel since it is reported in Nature Physics 19, 1286 (2023) in Fig.3.a. Then, I discussed in my previous report that the general ellipticity does not extract clear and reliable informations. The revised text has not changed my conclusion.

3) Regarding the critical magnetic field, I already stressed in the previous report that the authors greatly over-estimate the tunnel coefficient. Here, the moire potential depth is at least greater than 50 meV, given that the on-site repulsion is around 30 meV, so that the tunnel can not exceed 100 micro-eV, as shown in PRB 105, 165419. The authors have not calculated the tunnel coefficient, unlike suggested in my previous report. This parameter is crucial because I did its calculation, which agrees with the above reference, and strongly disagree with the estimation brought forward by the authors.

The parameter t^2/U is then not evaluated correctly. It is overestimated by at least 2 orders of magnitude, which rules out the validity of the authors discussion of Fig.5.

Conclusion: To my view this resubmission confirms that the authors report optically induced

polarisation of moire excitons, without new insights compared to Nature Physics 19, 1286 (2023).

Reviewer #3:

Remarks to the Author:

The revised manuscript is much improved. I believe the authors have satisfactorily addressed questions from me and the other two reviewers. The extensive revision and addition are more than adequate. I find the evidences of the proposed mechanism of exciton spin order convincing. This work adds to the growing body of work on magnetic order in moire systems. I recommend publication.

Reviewer #1 (Remarks to the Author):

The authors have answered convincingly to all my previous comments, they have furthermore improved the way the paper is written making it clearer for the reader. As I had said in my previous comments, the presented results are quite new and interesting and the very good agreement between the experimental data and simulation is convincing. In conclusion I think the paper deserve publication in Nature Communication.

We thank the reviewer for the positive assessment of our work and recommendation for publication.

Reviewer #2 (Remarks to the Author):

I have carefully read the revised manuscript from Xiong et al. together with the resubmission letter.

After this rebuttal I recommend not to publish this manuscript, which makes claims unsupported by experimental observations. In fact, I dispute the authors hypothesis and interpretations. Compared to the state-of-the-art (Nature Physics 19, 1286 (2023)) I find that this manuscript does not provide novel physics, and instead blurs the clear understanding given in Nature Physics 19, 1286 (2023).

We thank the reviewer for reviewing our paper again. Below we addressed all the concerns in a point-by-point manner.

1) The authors assume that excitons realise a Mott insulator at unity filling. This hypothesis lays the ground for all the interpretation. However, it is not verified and I dispute its validity: probe excitons either fill empty moire sites, or induce doublons. In Fig.1, around an average unity filling probe excitons still populate empty sites, as shown by line I, which by definition implies that the pump beam has not induced a Mott insulator, otherwise all sites would be occupied by one exciton.

We agree with the reviewer that the realization of a Mott insulator at unity filling lays the ground for all interpretations in this paper. As we shown in our previous study (*Science* 380,860-864(2023)) and detailed discussions in **Reply R1-3** below, our pump-probe technique directly measures the chemical potential of excitons and demonstrates an incompressible state of interlayer exciton at unity filling.

Reply R1: First, our observations contradict with the reviewer's claim that "In Fig.1, around an average unity filling probe excitons still populate empty sites" and instead are consistent with a Mott insulator of excitons, as detailed below.

In a Bose Hubbard model with strong on-site repulsion, excitons always occupy empty sites whenever available. Consequently, the probe-injected excitons should occupy empty

sites to form single-occupied sites at exciton filling $\nu_{\text{ex}} < 1$, and form doublons at $\nu_{\text{ex}} \geq 1$. This is exactly what we observed in the experiment: peak I (probe-induced emission from single-occupied site) exists at $\nu_{\text{ex}} < 1$ but not at $\nu_{\text{ex}} \geq 1$ (Fig. R1b). The sudden disappearance of peak I at $\nu_{\text{ex}} = 1$ indicates that the probe-injected excitons cannot find empty sites anymore, i.e., all sites are each occupied by one exciton. This is a hallmark of a Mott insulator of excitons. At $\nu_{\text{ex}} \geq 1$, the probe-injected excitons should only form doublons, whose emission has a higher energy due to the on-site repulsion between excision. This is also consistent with our observation that only peak II (emission from doublon sites) exists at $\nu_{\text{ex}} \geq 1$.

In short, our observations show that the probe-injected excitons occupy empty sites at $\nu_{\text{ex}} < 1$ and form doublons at $\nu_{\text{ex}} \geq 1$, which directly demonstrates a Mott insulator of excitons at $\nu_{\text{ex}} = 1$.

Fig. R1: Unpolarized pump-probe spectroscopy. **a**, Schematics of our pump-probe spectroscopy on a type-II hetero-bilayer WSe₂/WS₂. The background interlayer-exciton density (red) is controlled by the pump light and the charge density is kept at zero. The probe light injects an extra interlayer-exciton (blue), whose response is isolated through lock-in detection. **b**, Exciton-filling dependence of probe-induced PL spectrum using unpolarized pump and probe light. A sudden jump of exciton chemical potential at $\nu_{\text{ex}}=1$ is observed (white arrow), indicating an incompressible state of exciton. Low (high) energy emission peak is denoted as peak I (II).

In my previous report I already stressed that the intensity difference in the probe induced PL (Fig.1.b) while varying the pump beam power shows that the experimental approach does not allow to fully isolate the influence of the probe. Unfortunately, the authors have not replied to this concern that was very important because to my view it underlines that here the compressibility is not extracted.

Reply R2: In the previous around of report, the reviewer commented:

“... However, I am doubtful since in the panel b the amplitude of the probe PL strongly varies with the strength of the pump filling the lattice. Why is that? Why does the probe intensity increases until $\nu=1$ and then suddenly drops? ...”

We have extensively discussed in the last round of reply (Reply R5) why peak I intensity would suddenly drop at $\nu_{\text{ex}} = 1$. As also discussed above, this is because all sites are occupied by excitons and additional probe-induced excitons only form doublons. The sudden drop of peak I at $\nu_{\text{ex}} = 1$ directly confirms that our measurement fully isolates the probe-induced signal. Because the majority of the sites in the system are still single occupied from pump excitation, we would see strong peak I emission at $\nu_{\text{ex}} \geq 1$ if we were not able to completely isolate the probe influence.

We are also confused by the reviewer’s statement that “intensity difference in the probe induced PL while varying the pump beam power” indicates “the experimental approach does not allow to fully isolate the influence of the probe”. The fact that the probe-induced PL intensity changes with pump beam power simply indicates the existence of exciton-exciton interactions such that the density of the background excitons can affect the behaviors of the probe-induced excitons. The sudden jump of probe-induced PL from peak I to peak II at $\nu_{\text{ex}} = 1$ is a good example, which originates from the on-site repulsion between excitons. The intensity changes of peak I at $\nu_{\text{ex}} < 1$ can also be naturally understood from ordinary channels of exciton-exciton interaction such as density-dependent lifetime shortening from Auger recombination.

Moreover, the pump probe method introduced here does not allow to extract the chemical potential. It only provides the magnitude of on-site interactions, as already done in the manuscript by the Xu group (Nature Physics 19, 1286 (2023)) without probe beam. In the conclusion of the paper from Park et al. it is clearly stated that compressibility measurements are critical, and I agree with this. Such measurements are essential, as shown for instance in Nature 609, 485 (2022). Measuring exciton compressibility in the moire potential marks the next milestone in the field to my point of view.

Reply R3: At last, we take the opportunity to elaborate more on our technique. Our pump probe measurement is an optical analog of electrical capacitance measurements. The pump light tunes the background exciton density, and the weak probe light injects a small number of additional excitons, whose emissions are isolated through lock-in detection. This allows us to isolate the energy it takes to add/remove one (probe-induced) interlayer exciton from the system on top of a given (pump-controlled) background exciton density, which is by definition the exciton chemical potential. Experimentally, we find a sudden chemical potential jump of excitons at unity filling, which allows us to identify an incompressible state of interlayer excitons in this system.

As a comparison, in ultra-cold atom and quantum dot systems, local compressibility measurement is an established procedure to test a bosonic insulating state. Such measurement requires obtaining site-resolved particle population, which pairs well with these bottom-up systems due to both the relatively small system size and the large site separation. On the other hand, this procedure becomes inapplicable to condensed matter systems owing to key differences between the systems: the much larger number of particles and much smaller site separation in top-down systems make it impractical to resolve individual sites. Nevertheless, because condensed matter systems are typically in the thermodynamic limit, the compressibility κ can be directly obtained from its original definition $1/\kappa = n^2 d\mu/dn$ without involving fluctuation-dissipation theorem, where n is the particle density and μ is its chemical potential. Therefore, the well-established procedure to measure compressibility in condensed matter physics community is through

thermodynamic quantities, such as electrical capacitance. It is very rare to obtain compressibility of a top-down crystal by investigating behavior of individual sites within the system.

In this sense, our technique carries forward a well-established procedure in condensed matter physics community and applies it to a novel context of optical measurement and excitons. This is highly desirable since conventional optical measurements such as PL cannot obtain compressibility; while site-resolved measurement is technically too challenging given the small (<10 nm) site separation. Our technique can be further extended to flavor- and time-resolved measurements, thereby providing a powerful toolbox to investigate novel correlated states of bosons in condensed matter systems. Therefore, besides demonstrating new physics and opportunities in semiconducting moiré superlattices, we expect our work to benefit the community by introducing a (set of) novel and much-needed measurement techniques.

2) The authors present that increasing the moire filling enhances the optically induced exciton polarisation. However, this observation is not novel since it is reported in *Nature Physics* 19, 1286 (2023) in Fig.3.a. Then, I discussed in my previous report that the general ellipticity does not extract clear and reliable informations. The revised text has not changed my conclusion.

Reply R4: In *Nature Physics* 19, 1286 (2023) Fig. 3a, the authors reported increasing PL helicity of peak I with increasing circular pump power. Nevertheless, as we have explicitly emphasized in the last round of reply (Reply R10), the value of helicity at circular pump does not carry information of spin orders. Instead, our evidence of spin orders is non-trivial shapes in generalized helicity (GH), which is only analyzed in the present work. This can be readily seen from Fig. R2 a-b, which compares the PL raw helicity and GH for two different configurations: $\nu_{\text{ex}}=0.01$ at $T = 3$ K (blue color) and $\nu_{\text{ex}}= 1.3$ at $T = 60$ K (red color). In this example, the PL raw helicity at $\nu_{\text{ex}}= 1.3$ and $T = 60$ K is much larger than PL helicity at $\nu_{\text{ex}}=0.01$ at $T = 3$ K (0.5 vs 0.06 at circular pump). However, GH in both cases

is a constant, indicating that their difference in PL helicities is due to single particle effects and neither has exciton “spin” orders. As a contrasting example, Fig. R2 c-d compare the PL raw helicity and GH for $\nu_{\text{ex}}=1.1$ (magenta color) and 1.39 (orange color) at 3 K. The nontrivial shapes of GH indicate the existence of exciton “spin” orders. See Reply R10 in the last round for more details.

In summary, although the results of paper *Nature Physics* 19, 1286 (2023) could be affected the existence of spin orders, the approach used cannot probe the effect of spin orders. Therefore, the results, analysis and conclusions here are all fundamentally different from those in *Nature Physics* 19, 1286 (2023).

Fig. R2: Evidence of exciton spin orders from generalized helicity (GH). **a,b**, PL raw helicity η_{PL} (**a**) and GH (**b**) at $\nu_{\text{ex}}=0.01$, $T = 3$ K (blue color) and at $\nu_{\text{ex}}= 1.3$, $T = 60$ K (red color). GH is a constant in both cases. **c,d**, PL raw helicity η_{PL} (**c**) and GH (**d**) for $\nu_{\text{ex}}=1.1$ (magenta color) and 1.3 (orange color) at 3 K. The nontrivial shapes of GH indicate the existence of exciton “spin” orders.

3) Regarding the critical magnetic field, I already stressed in the previous report that the authors greatly over-estimate the tunnel coefficient. Here, the moire potential depth is at least greater than 50 meV, given that the on-site repulsion is around 30 meV, so that the tunnel can not exceed 100 micro-eV, as shown in PRB 105, 165419. The authors have not calculated the tunnel coefficient, unlike suggested in my previous report. This parameter is crucial because I did its calculation, which agrees with the above reference, and strongly disagree with the estimation brought forward by the authors. The parameter t^2/U is then not evaluated correctly. It is overestimated by at least 2 orders of magnitude, which rules out the validity of the authors discussion of Fig.5.

Reply R4: We thank the reviewer for pointing out related theoretical study *PRB* 105, 165419 and have included it in the revised manuscript as Ref. 36. In Fig. 6 of the reference, the calculated t_1/U_0 of $a^M \sim 8$ nm is at the order of 0.001, which gives hopping ~ 100 μ eV for $U_0 \sim 100$ meV. However, the system considered there is small-twist-angle MoSe₂/WSe₂, where exciton bands can be very different from the angle-aligned WS₂/WSe₂ studied in our work. It is also known that accurately predicting parameters for moiré materials remains a major theoretical challenge due to the complication from lattice relaxation and reconstruction in the system.

On the other hand, angle-aligned WS₂/WSe₂ system have been studied by several other works, and the hopping energy of excitons is consistently estimated to be several meV. For example, intralayer excitons in WSe₂/WS₂ has been studied experimentally in *Nature* 567, 76–80 (2019). The hopping energy is determined to be 1~2 meV (Fig. 4) under a moiré potential of amplitude ~ 100 meV. Since the intralayer and interlayers excitons have similar effective mass, interlayer excitons should also have hopping energy around 1~2 meV under a similar moiré potential. In addition, *Nature* 609, 52–57 (2022) uses large-scale GW-BSE calculations and determines the bandwidth W of exciton to be around 10 meV (Fig .1). The hopping energy of excitons is a fraction of W , thus several meV.

Furthermore, as we have emphasized in the last round of reply (Reply R12), a small hopping t of a few μ eV is directly contradicting our experimental results. First, if the

observed magnetic field dependence originates from field-induced spin polarization, as the reviewer proposed previously, changing the sign of h_{ext} would reverse the sign of spin polarization. In contrast, experimentally we observe symmetric field dependence of positive and negative h_{ext} in all measurements, thereby directly excluding a simple field-induced polarization. Second, we observe a saturation field of ~ 10 mT in magnetic field dependence, which corresponds to an energy scale of a few μeV , much smaller than our measurement temperature (3 Kelvin). Such sensitive magnetic field dependence is incompatible with a weak exchange interaction. If the exchange interaction were indeed a few μeV , as the reviewer proposed, the system would be effectively non-interacting at 3 K and the saturation field would be in the order of 1 T instead of 10 mT.

Reviewer #3 (Remarks to the Author):

The revised manuscript is much improved. I believe the authors have satisfactorily addressed questions from me and the other two reviewers. The extensive revision and addition are more than adequate. I find the evidences of the proposed mechanism of exciton spin order convincing. This work adds to the growing body of work on magnetic order in moire systems. I recommend publication.

We thank the reviewer for helping to improve the manuscript and recommendation for publication.

Reviewers' Comments:

Reviewer #2:

Remarks to the Author:

I regret to remain the only Reviewer disagreeing with the authors. Unfortunately I can not change my mind. The authors rely on qualitative argumentation to defend a strong claim, namely that they have emulated a "two-component Bose-Hubbard Hamiltonian". I continue to dispute this interpretation, for the same reasons, mainly 1) and 3) of my previous report. For 1), I argue that the technique used by the authors does not allow them to demonstrate, without any ambiguity, the buildup of a Mott insulating phase. This is the pre-requisite to possibly claim that the experiments obey the Bose-Hubbard model. For the reason 3), again the authors do not provide any quantitative evidence for the magnitude of the excitons tunnel strength t . Based on first principle calculation, in agreement with the literature, I evaluate that t is probably at least one order of magnitude smaller than what expected in the manuscript. This is a very serious issue ruling out the authors interpretation. In their reply, the authors mention a publication from the Berkeley groups and write that their measurements show that the tunnelling is large. I regret but the given reference does not quantify the tunnel strength in the moire lattice, while the experiments reported in Fig.5 do not suffice to quantify the tunnel strength.

Reply R1-R2:

I will try to make myself as clear and concise as possible:

The measurements displayed in Fig.1.c-d show that the probe induced PL has a low energy part, marking the fraction of empty sites, decreasing up to an average filling around 1.3 (the last point). At the same time, starting from filling above 0.5 the high-energy line develops and quantifies double occupancy. Unlike claimed by the authors, this observation implies that a mixture of regions with single, double occupancy, and empty sites, is probed throughout the explored range. The fraction of each contribution surely varies, but this is a classical regime very different from the Mott insulating regime.

Reply R3:

If the authors want to employ their method to measure the chemical potential at varying background density, and then deduce the system compressibility, then they must verify that the probe is indeed a probe. This implies that the probe intensity does not vary with the pump. Otherwise, additional physical processes enter and the measurements are not understood unless these processes are quantified. Well, this is exactly my question to the authors. The probe intensity decreases strongly while the authors increase the pump, and the authors do not provide quantitative measurements to interpret this behaviour. As a result, as previously stressed in my reports, I argue that the authors do not measure the exciton compressibility.

Reply R4:

I have carefully examined the Ref. Nature (2022) brought forward in the previous rebuttal. Indeed, it confirms the complexity of the excitons confinement in the moire potential. Nevertheless, the manuscript does not quantify the interlayer exciton tunnelling strength. Fig.1 only deals with intra-layer excitons. The work strongly insists on the difference between the regions where electrons and holes are confined. The band curvature of inter-layer excitons then remains to be quantified in order to confirm the authors argumentation. At the moment, first principle calculations lead to an order of magnitude that strongly contradicts the authors interpretation.

REVIEWER COMMENTS

Reviewer #2 (Remarks to the Author):

I regret to remain the only Reviewer disagreeing with the authors. Unfortunately I can not change my mind. The authors rely on qualitative argumentation to defend a strong claim, namely that they have emulated a “two-component Bose-Hubbard Hamiltonian”. I continue to dispute this interpretation, for the same reasons, mainly 1) and 3) of my previous report. For 1), I argue that the technique used by the authors does not allow them to demonstrate, without any ambiguity, the buildup of a Mott insulating phase. This is the pre-requisite to possibly claim that the experiments obey the Bose-Hubbard model. For the reason 3), again the authors do not provide any quantitative evidence for the magnitude of the excitons tunnel strength t . Based on first principle calculation, in agreement with the literature, I evaluate that t is probably at least one order of magnitude smaller than what expected in the manuscript. This is a very serious issue ruling out the authors interpretation. In their reply, the authors mention a publication from the Berkeley groups and write that their measurements show that the tunnelling is large. I regret but the given reference does not quantify the tunnel strength in the moire lattice, while the experiments reported in Fig.5 do not suffice to quantify the tunnel strength.

We thank the reviewer for his comments on the manuscript. Below we addressed all the concerns in a point-by-point manner.

Reply R1-R2:

I will try to make myself as clear and concise as possible:

The measurements displayed in Fig.1.c-d show that the probe induced PL has a low energy part, marking the fraction of empty sites, decreasing up to an average filling around 1.3 (the last point). At the same time, starting from filling above 0.5 the high-energy line develops and quantifies double occupancy. Unlike claimed by the authors, this observation implies that a mixture of regions with single, double occupancy, and empty sites, is probed throughout the explored range. The fraction of each contribution surely varies, but this is a classical regime very different from the Mott insulating regime.

Reply R1: We have used several different pump-probe configurations in Fig. 1 and 2 to access different information; and it seems that the reviewer is confused by the information and expectations from each configuration. In particular, the unpolarized pump-probe configuration directly measures exciton chemical potential (Fig. R1b, same as Fig. 1b in the main text). We expect peak I (emission from single-occupied site) to immediately disappear for $\nu_{\text{ex}} \geq 1$ since

probe-injected excitons can only create doublons at $\nu_{\text{ex}} \geq 1$. This is exactly what we have observed.

In contrast to Fig. 1b, Fig. 1c and 1d are probing spin-resolved responses through a different polarization configuration of the measurement. Peak I feature does persist to the highest measured fillings in both spin-up and spin-down detection channels. **However, this is entirely expected from the spin-1/2 Bose Hubbard model:** in spin-resolved measurement, peak I features are expected to persist up to $\nu_{\text{ex}} = 2$ because the formation of doublon is spin-selective and will have different effects on the spin-down and spin-up sites, as we have discussed in the original manuscript (line 103 to 114). The peak I signals from the two spin channels cancel at $\nu_{\text{ex}} \geq 1$ when summed up, again confirming that the probe only creates doublons at $\nu_{\text{ex}} \geq 1$. Therefore, the behaviors of peak I observed in all our measurements are consistent with expectations from an ideal Bose Hubbard model with a Mott insulator at $\nu_{\text{ex}} = 1$.

We agree with the reviewer that ideally peak II (emission from doublons) should only appear at $\nu_{\text{ex}} \geq 1$ in all measurement configurations. In Fig. 1c and 1d, peak II does emerge below $\nu_{\text{ex}}=1$. This is mainly due to the large intensity of the pulsed probe light we used for time-resolved measurements, which can transiently increase the exciton density by a significant amount to form doublons. Another important practical factor is sample inhomogeneity. Because exciton density depends on exciton lifetime, its spatial inhomogeneity is very sensitive to defects, strain etc. and is expected to be much larger than the charge case. At an average $\nu_{\text{ex}} < 1$, there could already be regions in the sample with $\nu_{\text{ex}} \geq 1$, which leads to doublon emission. These issues could be addressed or alleviated by using a weaker probe and/or a more homogeneous sample. For example, Fig. R1b uses a continuous wave (CW) probe with much smaller peak intensity. Therefore, peak II emerges only slightly below $\nu_{\text{ex}}=1$. In our previous work (Science 380,860-864 (2023)), we have used both very weak CW probe as well as a more homogeneous sample, therefore peak II emerges almost exactly at $\nu_{\text{ex}}=1$.

We have included the above discussions in Methods. We have also added the following discussions to the main text:

“... We note that while peak I is expected to disappear at $\nu_{\text{ex}} \geq 1$ in exciton chemical potential measurement, as is observed in Fig. 1b, it should persist in spin-resolved measurements until $\nu_{\text{ex}}=2$ (Fig. 1c and 1d) due to the spin-selective formation of doublons discussed above. This can also be understood by the cancellation of peak I signals when adding up the two spin channels at $\nu_{\text{ex}} \geq 1$ (Fig. 1c and 1d), which confirms that the probe light cannot add more single-occupied sites and can only create doublons. Meanwhile, peak II should ideally only emerge at $\nu_{\text{ex}} \geq 1$. The weak peak II features observed at $\nu_{\text{ex}} < 1$ in Fig. 1c and 1d are mainly due to the large intensity of the pulsed probe light used here for time-resolved measurements, as well as the spatial

inhomogeneity in the exciton density (see Methods: Discussion on doublon emission). These effects could be reduced by using a weaker probe and a more homogeneous sample, such as in Ref. ²⁵. A systematic study on probe intensity and sample homogeneity would help further quantify these effects.”

Fig. R1: Spin-1/2 Bose-Hubbard model a, Schematics of our pump-probe spectroscopy on a type-II hetero-bilayer WSe₂/WS₂. The background interlayer-exciton density (red) is controlled by the pump light and the charge density is kept at zero. The probe light injects an extra interlayer-exciton (blue), whose response is isolated through lock-in detection. **b**, Exciton-filling dependence of probe-induced PL spectrum using unpolarized pump and probe light. A sudden jump of exciton chemical potential at $\nu_{ex}=1$ is observed (white arrow), indicating an incompressible state of exciton. Low (high) energy emission peak is denoted as peak I (II). **c,d**, Polarization-resolved probe-induced PL spectra as a function of exciton filling. A linear pump light is used to generate equal numbers of *K* and *K'* valley excitons in the background, while an LCP probe light selectively excites extra *K* valley excitons. *K'* valley (**c**) or *K* valley (**d**) PL response induced by the probe light is collected separately. UKK (UKK') refers to pump injecting unpolarized excitons, probe injecting *K* valley excitons and PL detecting *K* (*K'*) valley excitons.

Reply R3:

If the authors want to employ their method to measure the chemical potential at varying

background density, and then deduce the system compressibility, then they must verify that the probe is indeed a probe. This implies that the probe intensity does not vary with the pump. Otherwise, additional physical processes enter and the measurements are not understood unless these processes are quantified. Well, this is exactly my question to the authors. The probe intensity decreases strongly while the authors increase the pump, and the authors do not provide quantitative measurements to interpret this behaviour. As a result, as previously stressed in my reports, I argue that the authors do not measure the exciton compressibility.

Reply R2: The pump and probe lights are completely independent from each other in our measurements. Therefore, the intensity of probe light does not vary with the pump light. Meanwhile, the probe-induced system response should naturally change with the pump intensity as long as there exists exciton-exciton interactions, as we detailed in the last round the review.

Reply R4:

I have carefully examined the Ref. Nature (2022) brought forward in the previous rebuttal. Indeed, it confirms the complexity of the excitons confinement in the moire potential. Nevertheless, the manuscript does not quantify the interlayer exciton tunnelling strength. Fig.1 only deals with intra-layer excitons. The work strongly insists on the difference between the regions where electrons and holes are confined. The band curvature of inter-layer excitons then remains to be quantified in order to confirm the authors argumentation. At the moment, first principle calculations lead to an order of magnitude that strongly contradicts the authors interpretation.

Reply R3: We agree with the reviewer that accurate prediction of parameters in the Bose Hubbard is complicated and has not been reported for the exact situation here. In the main text of revised manuscript, we have added explicit discussion of such complications and acknowledged the importance of future work, as detailed below:

“... While the phenomenological model here captures the salient features from the experiment, more quantitative theoretical studies are warranted to fully understand the exciton behaviors observed. The recent success of calculating moiré excitons from GW-BSE⁴⁸ could allow direct prediction of the exciton hopping in the Bose Hubbard model. Furthermore, additional effects not captured by the Bose Hubbard model, such as exciton dissociation and decoherence between the two valleys, could also contribute to the exotic exciton responses. Our results provide a valuable experimental reference for future theoretical studies.”